# KEAP1 loss modulates sensitivity to kinase targeted therapy in lung cancer

Elsa B Krall[1,2,3†‡], Belinda Wang[1,2,3†], Diana M Munoz[4†], Nina Ilic[1,2,3], Srivatsan Raghavan[1,2,3], Matthew J Niederst[4], Kristine Yu[4], David A Ruddy[4], Andrew J Aguirre[1,2,3], Jong Wook Kim[1,2,3], Amanda J Redig[1,3], Justin F Gainor[5], Juliet A Williams[4], John M Asara[6,7], John G Doench[2], Pasi A Janne[1,3], Alice T Shaw[5], Robert E McDonald III[4], Jeffrey A Engelman[4], Frank Stegmeier[4‡], Michael R Schlabach[4‡], William C Hahn[1,2,3*]

[1]Department of Medical Oncology, Dana-Farber Cancer Institute, Boston, United States; [2]Broad Institute of Harvard and MIT, Cambridge, United States; [3]Department of Medicine, Brigham and Women's Hospital and Harvard Medical School, Boston, United States; [4]Oncology Disease Area, Novartis Institute for Biomedical Research, Cambridge, United States; [5]Department of Medicine, Massachusetts General Hospital, Harvard Medical School, Boston, United States; [6]Department of Medicine, Beth Israel Deaconess Medical Center, Harvard Medical School, Boston, United States; [7]Cancer Center, Beth Israel Deaconess Medical Center, Harvard Medical School, Boston, United States

*For correspondence:
william_hahn@dfci.harvard.edu

†These authors contributed equally to this work

Present address: ‡KSQ Therapeutics, Cambridge, United States

**Abstract** Inhibitors that target the receptor tyrosine kinase (RTK)/Ras/mitogen-activated protein kinase (MAPK) pathway have led to clinical responses in lung and other cancers, but some patients fail to respond and in those that do resistance inevitably occurs (*Balak et al., 2006*; *Kosaka et al., 2006*; *Rudin et al., 2013*; *Wagle et al., 2011*). To understand intrinsic and acquired resistance to inhibition of MAPK signaling, we performed CRISPR-Cas9 gene deletion screens in the setting of BRAF, MEK, EGFR, and ALK inhibition. Loss of *KEAP1*, a negative regulator of NFE2L2/NRF2, modulated the response to BRAF, MEK, EGFR, and ALK inhibition in BRAF-, NRAS-, KRAS-, EGFR-, and ALK-mutant lung cancer cells. Treatment with inhibitors targeting the RTK/MAPK pathway increased reactive oxygen species (ROS) in cells with intact KEAP1, and loss of KEAP1 abrogated this increase. In addition, loss of KEAP1 altered cell metabolism to allow cells to proliferate in the absence of MAPK signaling. These observations suggest that alterations in the KEAP1/NRF2 pathway may promote survival in the presence of multiple inhibitors targeting the RTK/Ras/MAPK pathway.

## Introduction

The receptor tyrosine kinase (RTK)/ mitogen-activated protein kinase (MAPK) pathway plays an important role in the development of lung and other cancers, with the frequent occurrence of mutations or copy number alterations in multiple nodes of this pathway (*Ding et al., 2008*; *Imielinski et al., 2012*). However, single-agent therapy targeting this pathway has had limited clinical success. While BRAF and EGFR inhibitors can produce dramatic responses temporarily, acquired resistance inevitably occurs in lung and other cancers (*Balak et al., 2006*; *Kosaka et al., 2006*; *Rudin et al., 2013*; *Wagle et al., 2011*). In addition to this acquired resistance, many tumors also exhibit intrinsic resistance to these inhibitors (*Corcoran et al., 2012*; *Prahallad et al., 2012*), as well as to MEK inhibitors (*Sun et al., 2014*).

Several studies have now shown that a general theme of resistance to these targeted therapies is activation of the RTK/MAPK pathway by alternative mechanisms. For example feedback activation of EGFR has been shown to cause intrinsic resistance to BRAF inhibition in colon cancer (*Corcoran et al., 2012*; *Prahallad et al., 2012*), and resistance to BRAF inhibition in melanoma can be caused by reactivation of the MAPK pathway by RTKs, NRAS, or COT (*Johannessen et al., 2010*; *Nazarian et al., 2010*). In lung cancer, transcriptional induction of ERBB3 causes intrinsic resistance to MEK inhibition in KRAS-mutant cancers (*Sun et al., 2014*), and acquired resistance to EGFR inhibitors was found to result from amplification of MET (*Engelman et al., 2007*). These findings highlight the importance of maintaining RTK/MAPK signaling in lung and other cancers and also suggest redundancy among different genetic alterations in this pathway. Due to the many ways that cancers can acquire resistance to single therapies targeting the RTK/MAPK pathway, combination therapy may hold more promise for treating tumors with alterations in this pathway.

While patients with EGFR- and ALK-mutant lung cancer do often respond to EGFR and ALK inhibitors, these responses are temporary, as acquired resistance inevitably develops (*Tang et al., 2013*; *Wilson et al., 2015*). Likewise, in on-going clinical trials testing MEK and BRAF inhibitors in KRAS- and BRAF-mutant lung cancer, responses have been reported in some patients (*Blumenschein et al., 2015*; *Hyman et al., 2015*). However, it is clear that both intrinsic and acquired resistance limits efficacy of these inhibitors. To prospectively identify mechanisms of resistance to EGFR, ALK, BRAF, and MEK inhibition in lung cancer, we performed CRISPR-Cas9 drug resistance screens in five cancer cell lines with different alterations in the RTK/Ras/MAPK pathway. We identified a number of genes whose deletion confers cell survival in these contexts, and focused on KEAP1, as loss of this gene modulated the response to targeted therapies in multiple contexts.

## Results

### CRISPR-Cas9 screens identify genes whose loss confers resistance to MEK and BRAF inhibition

To identify genes that modulate the response to RTK/Ras/MAPK inhibition in lung cancer, we performed two sets of CRISPR-Cas9 knockout screens. In the first set (*Figure 1A*), we performed three genome scale screens with the MEK inhibitor trametinib, in the NRAS-mutant lung cancer cell line H1299 (NRAS$^{Q61K}$), the BRAF-mutant lung cancer cell line HCC364 (BRAF$^{V600E}$), and the KRAS-mutant lung cancer cell line CALU1 (KRAS$^{G12C}$). One additional screen was performed in HCC364 cells treated with the BRAF inhibitor vemurafenib. To perform genome scale screens, we introduced the GeCKO v2 library (*Shalem et al., 2014*) into Cas9-expressing cells, selected cells that incorporated the sgRNAs and allowed genome editing to occur over one week. Cells were then harvested for the Day 0 time point or passaged in the presence of trametinib or vemurafenib (*Figure 1A*). We used the lowest concentration of drug that inhibited ERK phosphorylation and resulted in proliferative arrest or death (*Figure 1—figure supplement 1*). Genomic DNA was isolated on Days 14 and 21, and sgRNAs were quantified by sequencing. sgRNAs that were enriched in the Day 14 and Day 21 samples compared to the Day 0 samples were then identified, using two methods: (1) a cutoff of log2 fold change of at least 2, or (2) a STARS score (*Doench et al., 2016*) of at least 5 (*Supplementary file 1*). Several of the genes that scored in these screens also scored in a previous vemurafenib resistance screen in BRAF-mutant melanoma (*Shalem et al., 2014*).

We annotated the functions of each of the genes that scored in these screens to determine if particular functional categories scored repeatedly (*Figure 1B*). As expected, several genes in the MAPK pathway scored, including NF1, a negative regulator of Ras/MAPK signaling, and DUSP1, a dual-specificity phosphatase that inhibits ERK. We also found several positive regulators of p38/JNK MAPK signaling, suggesting that these other MAPK pathways may play a pro-apoptotic or anti-proliferative role in these cells. PTEN, a negative regulator of PI3K/AKT signaling, and TSC1 and TSC2, negative regulators of mTOR signaling, also scored, suggesting that increased signaling through the PI3K/AKT/mTOR pathway compensates for loss of Ras/MAPK signaling. In addition to these expected pathways, several of the genes that scored are components of histone acetyltransferase (HAT) complexes or of the Mediator complex. There were also several genes whose products are components of E3 ubiquitin ligase complexes. Multiple transcription factors scored, as well as general transcription machinery genes. Other functional categories for which multiple genes scored

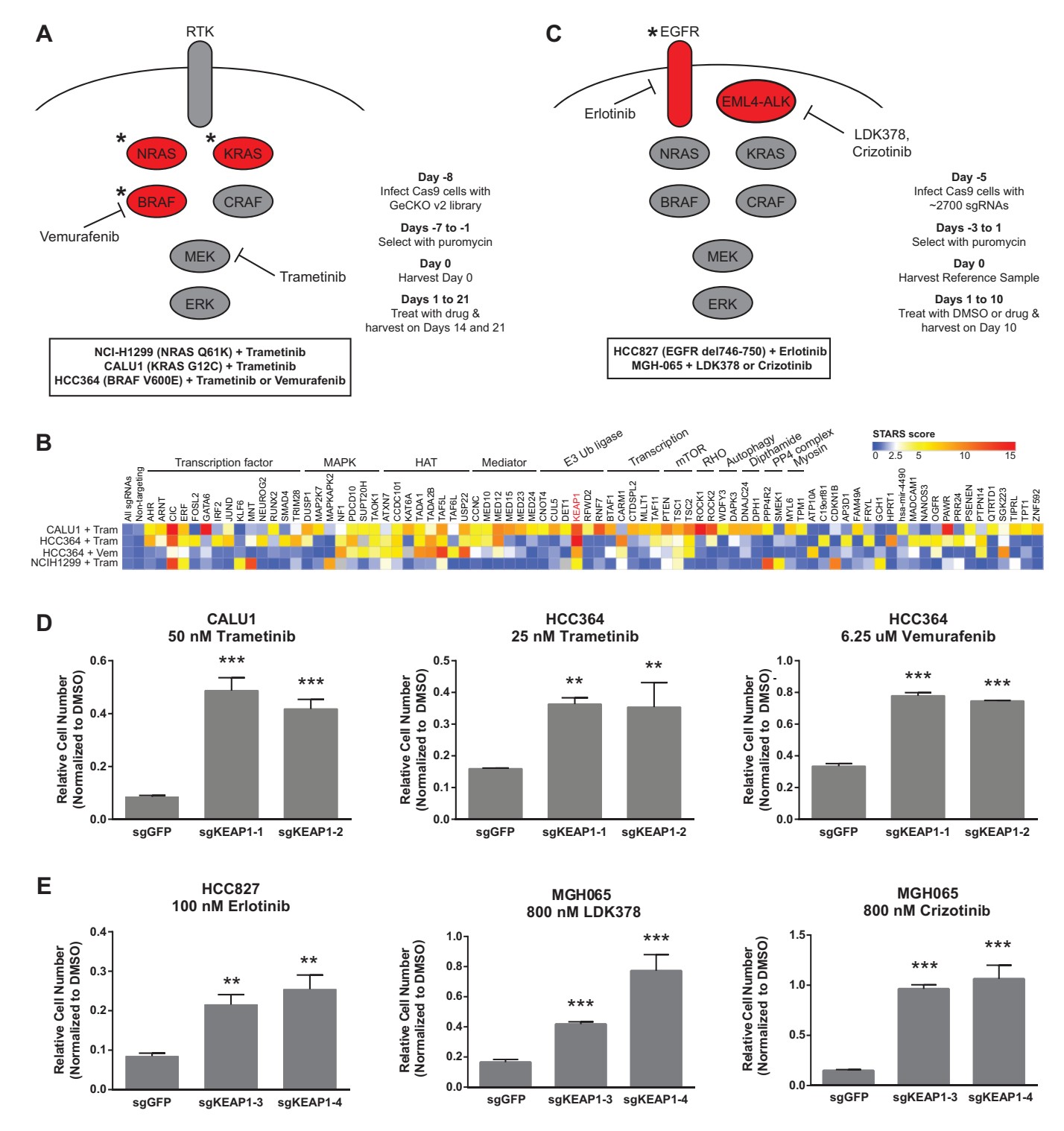

**Figure 1.** CRISPR-Cas9 genome scale drug resistance screens and validation that KEAP1-KO confers resistance. (A,C) Pathway schematics and screening timelines. (B) Heatmap showing the STARS score for each gene with a score of at least five in at least one screen. (D–E) Crystal violet colony formation assays. (D) 5000 CALU1 cells were treated with 50 nM trametinib for 17 days. 2000 HCC364 cells were treated with 25 nM trametinib or 6.25 uM vemurafenib for 21 days. (E) 10,000 HCC827 cells were treated with 100 nM erlotinib for 10 days. 10,000 MGH065 cells were treated with 800 nM LDK378 or 800 nM crizotinib for 10 days. Error bars represent the SD of the mean of triplicate wells. Experiments were performed two independent times and one representative experiment is shown. ***p=0.0001, **p<0.005.

*Figure 1 continued on next page*

*Figure 1 continued*

The following figure supplements are available for figure 1:

**Figure supplement 1.** Optimization of genome scale screening conditions.

**Figure supplement 2.** Optimization of focused sgRNA library screening conditions.

**Figure supplement 3.** EGFR and ALK inhibitor resistance screens.

**Figure supplement 4.** Immunoblots showing deletion of KEAP1 by sgRNAs in HCC364, CALU1, MGH-065, and HCC827.

**Figure supplement 5.** Crystal violet assays showing that KEAP1 loss confers resistance to afatinib and crizotinib.

**Figure supplement 6.** Erlotinib dose-response curves in HCC827.

include Rho signaling and histidine post-translational modifications. We noted that KEAP1, a substrate adaptor protein that targets NFE2L2/NRF2 for ubiquitination and proteasomal degradation, was the only gene that scored highly in all four genome scale screens (*Figure 1B* and *Supplementary file 1*).

The second set of screens (*Figure 1C*) were run independently using a more focused sgRNA library comprised of druggable targets, including kinases, proteases, phosphatases, and ligases (*Munoz et al., 2016*). This library was used to screen HCC827 cells (EGFR$^{\Delta746-750}$) treated with the EGFR inhibitor erlotinib, and the patient-derived ALK-mutant line MGH-065 treated with the ALK inhibitors crizotinib or LDK-378 (*Figure 1—figure supplement 2*). Similar to the genome scale screens described above, Cas9-expressing cells were infected with the sgRNA library, selected for 4 days and cultured for an additional 10 days in the presence of DMSO or drug. SgRNAs that were enriched in the Day 10 drug-treated samples compared to the DMSO-treated samples were then identified. Notably, loss of KEAP1 also promoted survival in the presence of EGFR and ALK inhibitors (*Figure 1—figure supplement 3* and *Supplementary file 2*). Thus, loss of KEAP1 was found to modulate the response to multiple targeted therapeutics in different genetic contexts using two independent sgRNA libraries.

## KEAP1$^{KO}$ confers resistance through increased NRF2 activity

To validate that KEAP1 loss modulates sensitivity to multiple inhibitors, we infected HCC364 (BRAF$^{V600E}$), CALU1 (KRAS$^{G12C}$), HCC827 (EGFR$^{\Delta746-750}$), and MGH-065 (EML4-ALK) cells with sgRNAs targeting KEAP1 or GFP (*Figure 1—figure supplement 4*). Cells were then seeded at low density in 12- or 24-well plates and treated with drug. Cell viability was assessed by crystal violet staining. Deletion of KEAP1 (KEAP1$^{KO}$) decreased sensitivity to each of the inhibitors that were tested (*Figure 1D–E*). In addition to the lines used in the CRISPR-Cas9 screens, we also tested the ability of KEAP1 loss to modulate sensitivity to EGFR or ALK inhibition in two additional cell lines. KEAP1$^{KO}$ decreased sensitivity to afatinib treatment in NCI-H1975 (EGFR$^{L858R/T790M}$) cells and to crizotinib treatment in NCI-H3122 (EML4-ALK) cells (*Figure 1—figure supplement 5*). In short-term dose response experiments, loss of KEAP1 also caused a shift in the IC50 of erlotinib in HCC827 cells (*Figure 1—figure supplement 6*).

Unlike many other reported mechanisms of resistance to inhibitors of the RTK/Ras/MAPK pathway (*Rudin et al., 2013*; *Wagle et al., 2011*; *Corcoran et al., 2012*; *Prahallad et al., 2012*; *Sun et al., 2014*; *Johannessen et al., 2010*; *Nazarian et al., 2010*), we found that KEAP1$^{KO}$ did not restore ERK activation (*Figure 2A* and *Figure 2—figure supplement 1A*), indicating that KEAP1$^{KO}$ does not decrease sensitivity by reactivating the MAPK pathway. Loss of KEAP1 also did not affect expression of the pro-apoptotic protein BIM (*Figure 2—figure supplement 1B*). We concluded that loss of KEAP1 did not re-activate MAPK signaling or generally promote survival by inhibiting apoptosis.

KEAP1 serves as a substrate adaptor protein that recruits the CUL3 ubiquitin ligase to NRF2, targeting it for proteasomal degradation (*Kobayashi et al., 2004*). As expected, we found that KEAP1$^{KO}$ led to increased NRF2 protein levels (*Figure 2B* and *Figure 2—figure supplement 1C*).

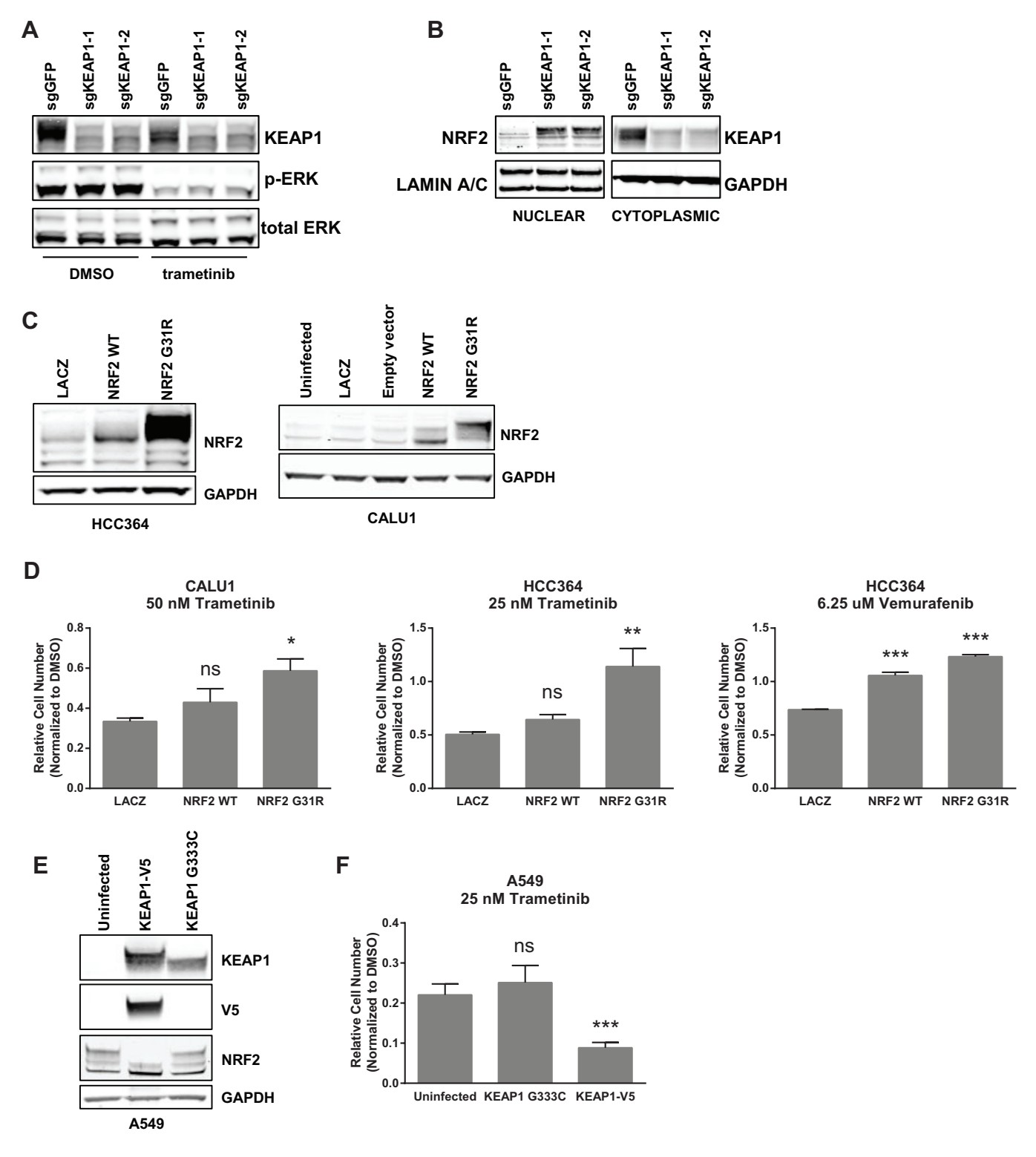

**Figure 2.** KEAP1-KO alters NRF2 levels. (**A**) KEAP1-KO does not affect p-ERK. Whole cell lysates of HCC364-Cas9 cells with the indicated sgRNAs treated with DMSO or trametinib for 48 hr. (**B**) KEAP1-KO increases NRF2 levels. Nuclear and cytoplasmic fractions of HCC364 cells. (**C**) Immunoblot showing NRF2 expression in CALU1 and HCC364. (**D**) Crystal violet colony formation assays. 10,000 CALU1 cells expressing the indicated ORFs were treated with DMSO for 8 days or trametinib for 10 Days. 10,000 HCC364 cells expressing the indicated ORFs were treated with DMSO for 10 days or trametinib/vemurafenib for 21 days. Error bars represent the SD of the mean of triplicate wells. (**E**) Immunoblot showing expression of wildtype KEAP1
*Figure 2 continued on next page*

*Figure 2 continued*
or KEAP1 G333C in A549 cells. (**F**) Expression of wildtype KEAP1 resensitized A549 cells to trametinib. 5000 cells were treated with 25 nM trametinib for 12 days. Error bars represent the SD of six wells. (**A–F**) Experiments were performed two independent times, and one representative experiment is shown. \*\*\*p=0.0001, \*\*p=0.0005, \*p<0.002.
The following figure supplements are available for figure 2:
**Figure supplement 1.** KEAP1-KO alters NRF2 levels and does not activate ERK.
**Figure supplement 2.** NRF2 is necessary and sufficient for resistance.

To determine if increased NRF2 levels were necessary for KEAP1 loss to modulate drug sensitivity, we knocked out both KEAP1 and NRF2 in HCC827 cells. KEAP1 loss no longer promoted survival when NRF2 was also lost (*Figure 2—figure supplement 2A*). To determine if increased NRF2 levels were sufficient to promote survival, we overexpressed wildtype NRF2 or two different NRF2 mutants (G31R or D29H), which contain mutations in the KEAP1 binding domain. Overexpression of wildtype NRF2 led to an increase in cell survival in the presence of multiple drugs. Overexpression of NRF2$^{G31R}$ or NRF2$^{D29H}$ resulted in higher NRF2 protein levels and increased resistance to trametinib, vemurafenib, and erlotinib (*Figure 2D* and *Figure 2—figure supplement 2B*), suggesting that elevated NRF2 levels in KEAP1$^{KO}$ cells mediates resistance. Although CALU1 cells harbor a KEAP1$^{P128L}$ mutation, this mutation has not been reported in the cBioPortal or COSMIC (*Cerami et al., 2012*; *Gao et al., 2013*; *Forbes et al., 2015*), and NRF2 levels increased upon KEAP1 knockout (*Figure 2—figure supplement 1*), suggesting that the regulation of NRF2 by KEAP1 is intact in these cells. We also found that restoring wildtype KEAP1 expression in A549 cells, which are KRAS-mutant and KEAP1-null, increased their sensitivity to trametinib. In contrast, expression of the KEAP1$^{G333C}$ mutant, which does not regulate NRF2, failed to alter trametinib sensitivity (*Figure 2E,F*). Together these observations suggest that increased NRF2 levels upon loss of KEAP1 modulates sensitivity to therapies targeting the RTK/MAPK pathway.

Loss of KEAP1 has previously been reported to confer resistance to several chemotherapeutics (*Ohta et al., 2008*; *Shibata et al., 2008a*; *Wang et al., 2008*; *Zhang et al., 2010*). However, it is also clear that KEAP1/NRF2 and the MAPK pathway are mechanistically linked (*DeNicola et al., 2011*; *Sun et al., 2009*). To further explore the mechanism by which KEAP1$^{KO}$ promotes survival in the presence of RTK/MAPK inhibitors, we investigated whether drug treatment affected KEAP1/NRF2 signaling. A prior report demonstrated that Ras/MAPK/Jun signaling increased expression of NRF2 mRNA and NRF2 target genes (*DeNicola et al., 2011*), so we hypothesized that drug treatment would decrease expression of NRF2 mRNA and NRF2-regulated target genes. Surprisingly, we found that trametinib, erlotinib, and LDK-378 treatment increased rather than decreased expression of NRF2 mRNA and NRF2 target genes (*Figure 3A* and *Figure 3—figure supplement 1*) in HCC364, CALU1, MGH-065, and HCC827 cells. As expected, KEAP1$^{KO}$ also increased NRF2 target gene expression. Trametinib treatment also increased NRF2 protein levels and induced a shift in the migration of NRF2 protein on SDS-PAGE, whereas KEAP1$^{KO}$ maintained the expression of the higher molecular weight form of NRF2 (*Figure 3B* and *Figure 3—figure supplement 2*). Although trametinib treatment decreased KEAP1 expression and increased NRF2 expression, the combination of KEAP1 knockout with trametinib treatment caused a greater effect on KEAP1 levels, NRF2 levels, and NRF2 target gene expression. These observations suggest that treatment with these inhibitors partially activates NRF2, and that loss of KEAP1 leads to further NRF2 activation, which decreases drug sensitivity.

## Loss of KEAP1 promotes survival by increasing glutathione synthesis and decreasing drug-induced ROS

The KEAP1/NRF2 axis responds to oxidative and electrophilic stress by regulating expression of drug efflux pumps, by scavenging reactive oxygen species (ROS), and by altering cell metabolism (*Hayes and Dinkova-Kostova, 2014*). We investigated whether each of these functions was involved in modulating sensitivity to RTK/MAPK inhibition. Since MAPK pathway inhibition is maintained in KEAP1$^{KO}$ cells (*Figure 2A*), drug efflux likely does not explain resistance. To determine if loss of

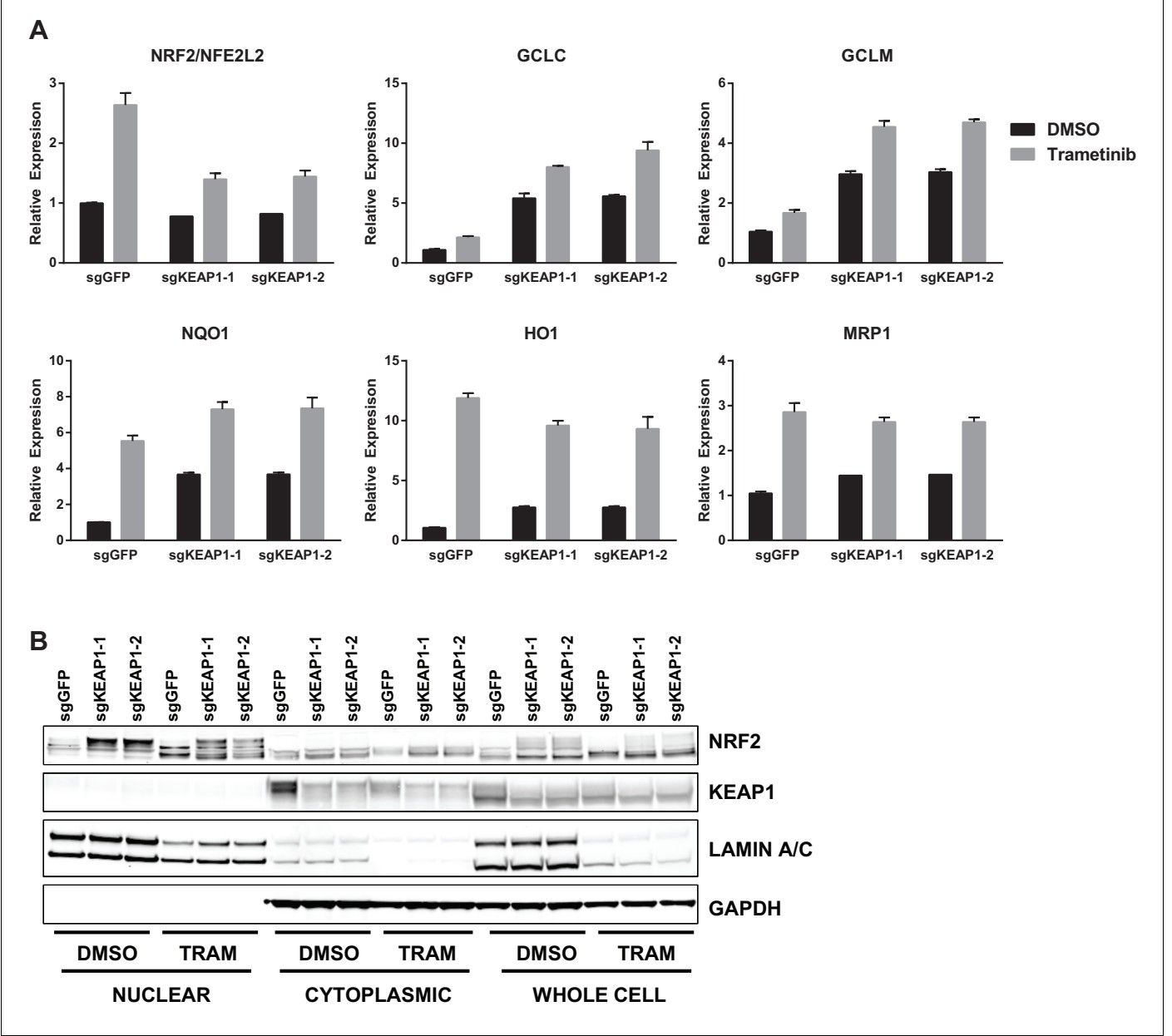

**Figure 3.** Trametinib treatment and KEAP1-KO increase NRF2 activity. (**A**) Expression of NFE2L2/NRF2 mRNA and NRF2 target genes in HCC364 treated with DMSO or trametinib for 72 hr. Error bars represent the SD of the mean of three biological replicates. (**B**) HCC364 cells treated with DMSO or trametinib for 72 hr. TRAM, trametinib. (**A**,**B**) Experiments were performed two independent times, and one representative experiment is shown.

The following figure supplements are available for figure 3:

**Figure supplement 1.** Expression of NFE2L2/NRF2 mRNA and NRF2 target genes in (**A**) CALU1 cells, (**B**) HCC827 cells, or (**C**) MGH-065 cells treated with DMSO or the indicated drug for 72 hr.

**Figure supplement 2.** CALU1 cells treated with DMSO or trametinib for 72 hr.

KEAP1 promotes survival through modulation of ROS, we treated control or KEAP1$^{KO}$ cells with DMSO or drug and measured ROS. We found that treatment with trametinib, erlotinib, or LDK-378 induced ROS in KEAP1-intact cells and that ROS was dramatically decreased in KEAP1$^{KO}$ cells or NRF2 overexpressing cells (**Figure 4A** and **Figure 4—figure supplement 1**), suggesting that

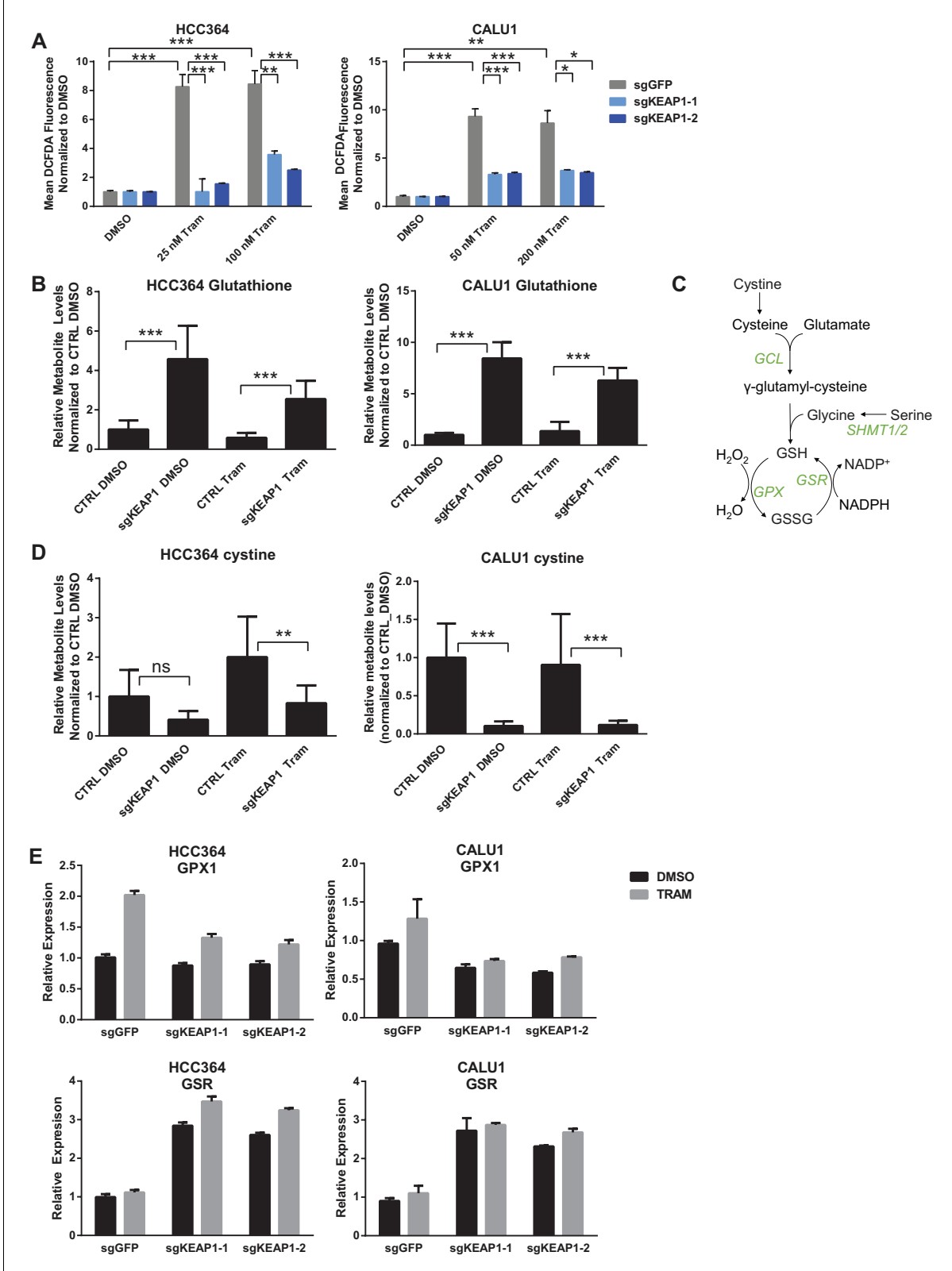

**Figure 4.** KEAP1-KO reduces trametinib-induced ROS through increased glutathione synthesis. (**A**) HCC364 or CALU1 cells were treated with DMSO or trametinib for 72 hr. ROS was measured by DCFDA fluorescence. Error bars represent SD of the mean of two biological replicates. (**B, D**) CALU1 or HCC364 cells were treated with DMSO or trametinib for 72 hr and metabolites were measured by liquid chromatography tandem mass spectrometry.
*Figure 4 continued on next page*

*Figure 4 continued*

Error bars represent SD of eight replicates. (C) Glutathione synthesis pathway. (E) Expression of GPX1 and GSR in HCC364 and CALU1 cells treated with DMSO or trametinib for 72 hr. Error bars represent the SD of the mean of three biological replicates. ***p<0.005, **p<0.01, *p<0.02.

The following figure supplements are available for figure 4:

**Figure supplement 1.** KEAP1-KO reduces drug-induced ROS.

**Figure supplement 2.** Glutathione levels in HCC364 and CALU1 cells.

**Figure supplement 3.** KEAP1-KO alters expression of SHMT1/2 in some cell lines.

**Figure supplement 4.** KEAP1-KO alters expression of GPX1 and GSR.

KEAP1$^{KO}$ may promote survival by reducing ROS levels. The ability of RTK/MAPK inhibitors to induce ROS in control cells may explain their ability to increase NRF2 levels, as increased ROS would lead to release of NRF2 from KEAP1 regulation (*Kobayashi et al., 2004*). Further activation of NRF2 through deletion of KEAP1 may then reduce ROS levels, leading to increased survival in the presence of drug.

To better understand how KEAP1$^{KO}$ reduced ROS, we performed metabolite profiling in control or KEAP1$^{KO}$ CALU1 and HCC364 cells treated with DMSO or trametinib for 72 hr. In both cell lines, the most highly increased metabolite in KEAP1$^{KO}$ cells treated with trametinib compared to control cells treated with trametinib was reduced glutathione (GSH), an antioxidant (*Figure 4B* and *Supplementary file 3*). In addition to glutathione, we identified several other changes in the glutathione synthesis pathway (*Figure 4C*). Both the reduced (GSH) and oxidized (GSSG) form of glutathione were increased in KEAP1$^{KO}$ cells, although the ratio of reduced to oxidized glutathione (GSH/GSSG) also increased (*Figure 4—figure supplement 2*), indicating a more reduced intracellular environment. We also found that cystine, the oxidized form of cysteine, was among the most decreased metabolites in KEAP1$^{KO}$ cells compared to control cells (*Figure 4D*), suggesting that KEAP1$^{KO}$ contributes to a change in oxidative state in which cystine is converted to cysteine, which is then used to make glutathione.

In addition to the changes in metabolite levels, we also observed several changes in gene expression that were consistent with increased glutathione synthesis in KEAP1$^{KO}$ cells. Glutamate cysteine ligase (GCL) catalyzes the rate-limiting step in glutathione synthesis, and both subunits of this enzyme (GCLC and GCLM) are NRF2 target genes. As described above, expression of both of these genes was higher in drug-treated KEAP1$^{KO}$ cells compared to drug-treated control cells (*Figure 3A* and *Figure 3—figure supplement 1*). Glycine is also a precursor to glutathione, and NRF2 has previously been shown to regulate expression of serine hydroxymethyltransferase (SHMT1,2) (*DeNicola et al., 2015*), which catalyzes the conversion of serine to glycine. We found that expression of SHMT1 and SHMT2 decreased upon trametinib treatment in CALU1 and HCC364 control cells but was partially maintained in KEAP1$^{KO}$ cells, although this was not observed in MGH-065 and HCC827 cells (*Figure 4—figure supplement 3*). In addition, we found that expression of glutathione peroxidase (GPX1), which converts GSH to GSSG, increased with drug treatment in control cells, but this increase was abrogated in KEAP1$^{KO}$ cells (*Figure 4E* and *Figure 4—figure supplement 4*). Conversely, expression of glutathione reductase (GSR), which converts GSSG to GSH, was higher in KEAP1$^{KO}$ cells than in control cells (*Figure 4E* and *Figure 4—figure supplement 4*). These observations suggest that the conversion of oxidized to reduced glutathione may be involved in the modulation of drug sensitivity meditated by loss of KEAP1.

To determine whether the increased glutathione levels promoted survival in the presence of drug, we treated KEAP1-intact cells with trametinib and N-acetyl cysteine (NAC), an antioxidant and glutathione precursor. NAC reduced ROS and decreased sensitivity to trametinib (*Figure 5A,B*), indicating that ROS reduction through increased glutathione synthesis is important for KEAP1$^{KO}$-mediated survival. To further investigate whether glutathione synthesis and ROS reduction was important for survival, we treated cells with trametinib and buthionine sulfoximine (BSO), which inhibits GCL. The combination of BSO and trametinib increased ROS and greatly decreased viability

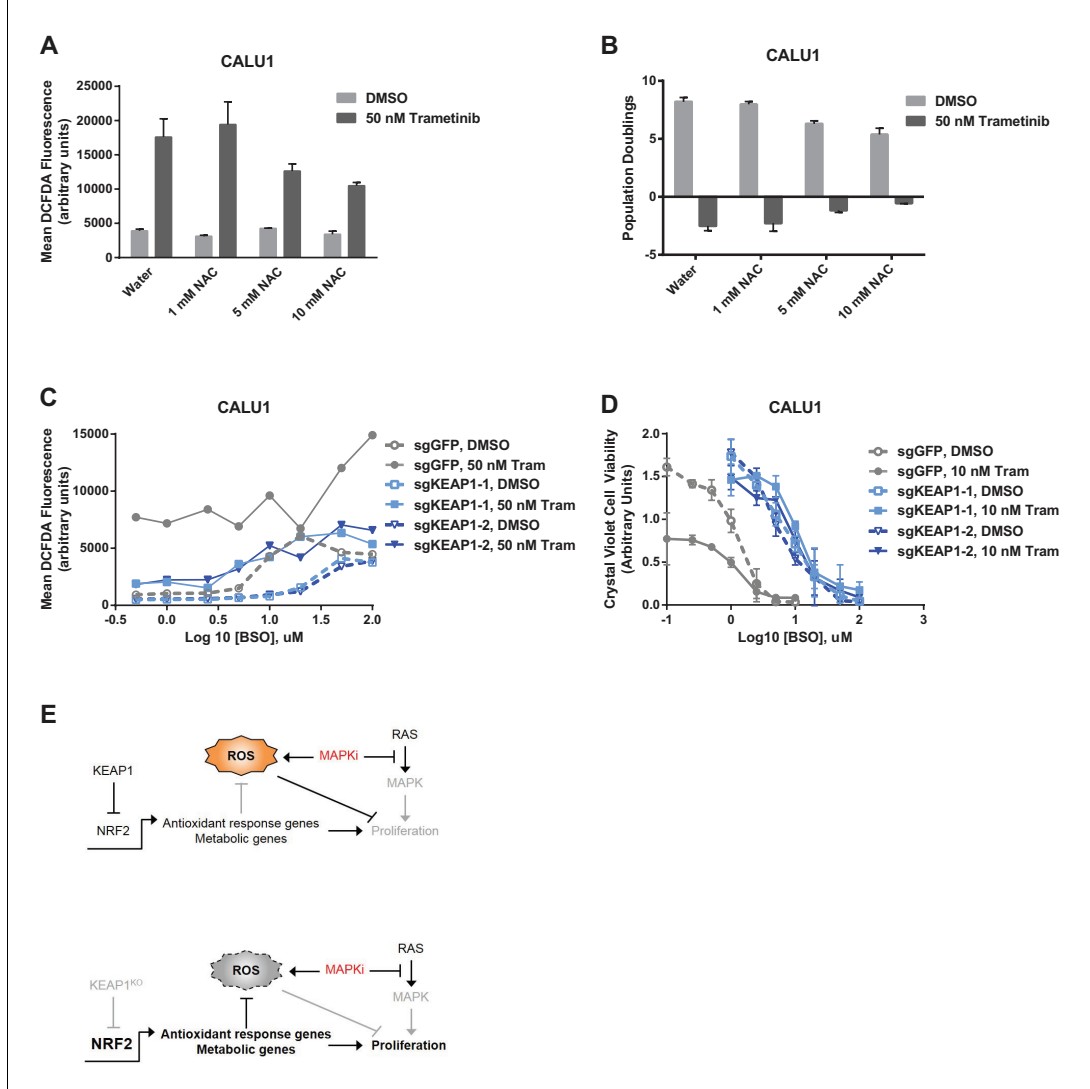

**Figure 5.** Altering ROS levels affects cell survival. (**A**, **B**) CALU1 cells were treated with DMSO or 50 nM trametinib and the indicated concentration of N-acetyl cysteine (NAC) for 16 days. (**A**) ROS was measured by DCFDA assay. Error bars represent SD of two replicates. (**B**) Population doublings of trametinib-treated cells compared to DMSO-treated cells. Error bars represent SD of two replicates. (**C**) KEAP1-KO reduces trametinib- and BSO-induced ROS in CALU1 cells. Cells were treated for 72 hr, and ROS was measured by DCFDA assay. Error bars represent SD of two replicates. (**D**) KEAP1-KO increases cell viability in CALU1 cells treated with trametinib and BSO. 20,000 CALU1 cells were treated with DMSO plus BSO for 7 days or trametinib plus BSO for 12 days. Error bars represent SD of triplicate wells. (**E**) Model of how KEAP1-KO confers resistance to trametinib. MAPK pathway inhibitors turn off MAPK signaling and also induce ROS, leading to low level activation of NRF2. Upon KEAP1 loss, NRF2 activity further increases, altering expression of genes involved in the antioxidant response and metabolism, allowing for proliferation in the absence of MAPK signaling.

The following figure supplements are available for figure 5:

**Figure supplement 1.** KEAP1-KO reduces ROS and increases viability in the presence of BSO.

**Figure supplement 2.** Expression of WT KEAP1 but not G333C in KEAP1-null A549 cells increases trametinib- and BSO-induced ROS.

**Figure supplement 3.** KEAP1-KO alters cell metabolism.

in control cells expressing sgGFP, while KEAP1[KO] prevented the BSO-induced decrease in viability (*Figure 5C,D* and *Figure 5—figure supplement 1*). Furthermore, combined treatment with BSO and trametinib dramatically increased ROS levels in A549 cells in which wildtype KEAP1 expression had been restored, but not in the parental cells or cells expressing KEAP1[G333C] (*Figure 5—figure supplement 2*). These observations indicate that KEAP1[KO] can decrease ROS even when GCL is inhibited, suggesting that the conversion of oxidized to reduced glutathione may be more important for promoting survival in the presence of drug. Together, these observations suggest that trametinib treatment induces ROS, which activates NRF2 to low levels. Loss of KEAP1 leads to further activation of NRF2, which decreases drug sensitivity in part by increasing glutathione and decreasing ROS.

In addition to regulating ROS, NRF2 has been reported to regulate the expression of metabolic genes (*DeNicola et al., 2015*; *Mitsuishi et al., 2012*). We found that glycolytic and pentose phosphate pathway intermediates were higher in trametinib-treated KEAP1[KO] cells compared to trametinib-treated control cells (*Supplementary file 3*), indicating that KEAP1 knockout partially restores anabolic metabolism. We also found that expression of several NRF2 target genes involved in the pentose phosphate pathway, de novo nucleotide synthesis, and NADPH synthesis was higher in drug-treated KEAP1[KO] cells than in drug-treated control cells (*Figure 5—figure supplement 3*). Together these observations support a model in which RTK/MAPK pathway inhibitors block MAPK signaling and induce ROS, which activates NRF2 to low levels. KEAP1 loss increases NRF2 activity, which reduces ROS through increased glutathione synthesis and alters cell metabolism, allowing cells to proliferate in the absence of MAPK signaling (*Figure 5E*).

To determine whether KEAP1 loss also conferred resistance in vivo, we performed a xenograft study, in which HCC827 KEAP1[KO] or control cells were implanted subcutaneously. When the average tumor volume reached 100–200 mm[3], mice were randomized to three groups: vehicle, 5 mg/kg erlotinib, or 12 mg/kg erlotinib. KEAP1[KO] cells showed a proliferative advantage over control cells in both the presence and absence of erlotinib (*Figure 6*).

## Discussion

Previous genome scale screens to identify mechanisms of resistance to targeted therapeutics have focused on a single genetic alteration (*Shalem et al., 2014*; *Berns et al., 2007*; *Johannessen et al.,*

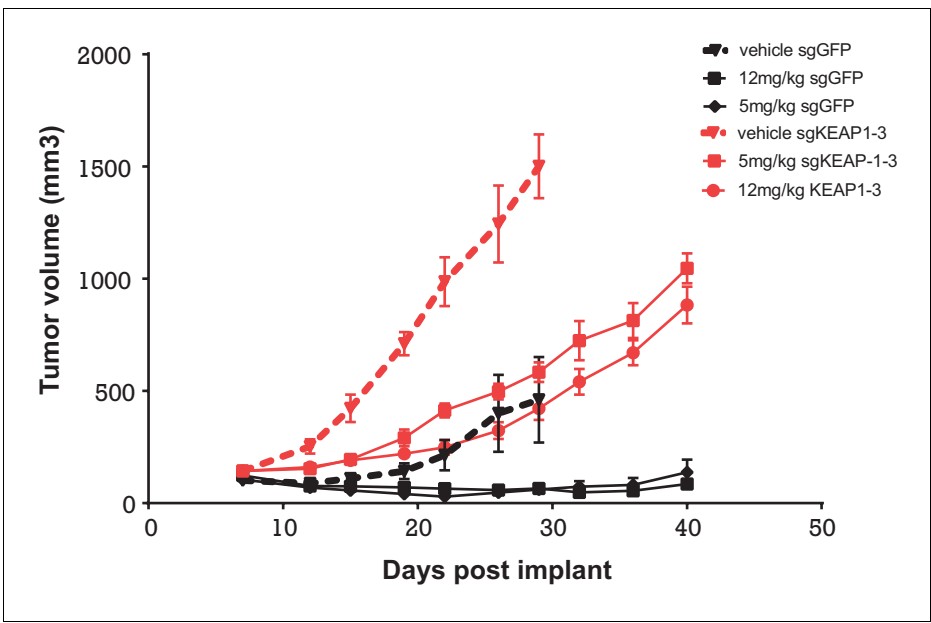

**Figure 6.** Alterations in the KEAP1/NRF2 pathway alter tumor growth in vivo. Growth of HCC827 sgGFP and sgKEAP1 cells in vivo. HCC827 sgGFP or sgKEAP1 cells were implanted bilaterally and treatment was initiated when average tumor volume reached 100–200 mm3. Data represent mean ± SD (n = 12).

2013; *Whittaker et al., 2013*). Here we have expanded this approach to identify mechanisms of resistance that are shared across multiple targeted therapeutics in several genetic contexts. We have found that KEAP1 loss modulates sensitivity to inhibition of EGFR, ALK, BRAF, or MEK in lung cancer cell lines with EGFR, ALK, BRAF, KRAS, or NRAS mutations. In multiple independent CRISPR-Cas9 screens, we found that sgRNAs targeting KEAP1 were more highly enriched than sgRNAs targeting NF1, DUSP1, PTEN, or TSC1/2, known negative regulators of the MAPK and PI3K pathways. Previous studies using shRNA screens to identify potential combination therapies for Ras-mutant cancers found that RTK/MAPK pathway reactivation (*Manchado et al., 2016*) or induction of anti-apoptotic proteins (*Corcoran et al., 2013*) limits therapeutic efficacy. Importantly, loss of KEAP1 does not lead to reactivation of the MAPK pathway or decreased levels of the pro-apoptotic protein BIM, and instead reduces drug-induced ROS and alters cell metabolism.

KEAP1 loss or NRF2 overexpression is sufficient to restore cell proliferation in the absence of MAPK signaling. It has previously been shown that ROS detoxification and modulation of redox state by NRF2 contributes to tumorigenesis (*DeNicola et al., 2011*), and NRF2 has recently been found to be an oncogene (*Kim et al., 2016*). In addition to its role in regulating redox state, NRF2 has also previously been shown to regulate expression of many metabolic enzymes, redirecting glucose and glutamine into anabolic pathways that support proliferation (*Mitsuishi et al., 2012*) and controlling serine and glycine biosynthesis to support glutathione and nucleotide production (*DeNicola et al., 2015*). Thus, increased expression of NRF2 upon KEAP1 loss can confer resistance to RTK/MAPK pathway inhibition by reducing ROS and regulating metabolic pathways. Chio *et al.* recently reported that NRF2 supports pancreatic tumor maintenance, and that combined targeting of AKT and glutathione synthesis inhibits pancreatic cancer (*Chio et al., 2016*). While their study focused on the role of NRF2 in regulating mRNA translation in pancreatic cancer, their findings regarding glutathione synthesis being a key function of NRF2 are in concordance with our observations.

We recently found that the number of CRISPR/Cas9-induced DNA breaks dictates a gene-independent anti-proliferative response in cells, such that targeting amplified regions decreases viability (*Munoz et al., 2016*; *Aguirre et al., 2016*). This effect confounds the use of CRISPR/Cas9 negative selection screening to identify essential genes in amplified regions. We do not believe that this effect is relevant to this study, in which we have performed positive selection screens to identify genes whose loss promotes proliferation under drug treatment. Moreover, we directly compare the same cells under two conditions; thus, any genes that are affected by the gene-independent effect will score in both conditions.

A recent vemurafenib BRAF$^{V600E}$ basket trial showed that 42% of lung cancers with the BRAF V600E mutation responded to vemurafenib (*Hyman et al., 2015*). As seen with vemurafenib treatment in melanoma or with EGFR inhibitors in lung cancer, acquired resistance will likely arise. Furthermore, while MEK inhibitors only elicit responses in a small number of lung cancer patients (*Blumenschein et al., 2015*), these responders are also likely to develop resistance. Predicting how resistance may arise in these patients will be important for developing more effective combination therapies. In addition, for those patients that do not initially respond, intrinsic resistance in a subset of these patients may be explained by the mechanisms we describe here. The KEAP1/NRF2 pathway is genetically altered in approximately 30% of lung squamous cell carcinomas and approximately 20% of lung adenocarcinomas. Alterations in this pathway can co-occur with alterations in the RTK/Ras pathway (*Cerami et al., 2012*; *Gao et al., 2013*; *Cancer Genome Atlas Research Network, 2014*), although KEAP1/NRF2 alterations are enriched in the 'oncogene negative' subset of lung cancers (*Cancer Genome Atlas Research Network, 2014*). BRAF and MEK inhibitors are currently being tested in clinical trials for RAS- and BRAF-mutant lung cancer. However, for most of these trials matched pre-treatment and post-relapse biopsy specimens are not available for molecular analysis of resistance mechanisms. Gainor *et al.* recently identified a NRF2 mutation in a patient with acquired resistance to an ALK inhibitor (*Gainor et al., 2016*). This mutation (E79Q) is in a mutational hotspot and has previously been shown to impair recognition of NRF2 by KEAP1, thus activating the pathway (*Shibata et al., 2008b*). This tumor also harbored a secondary ALK mutation of unknown function and became resistant to a second generation ALK inhibitor. Thus it is possible that the NRF2 mutation contributed to survival in the presence of crizotinib treatment and allowed the cells to acquire additional resistance mutations over time.

Although KEAP1/NRF2 alterations are known to confer resistance to chemotherapy, KEAP1/NRF2 mutation status is not used to make treatment decisions in lung cancer. As more patients are treated

with RTK/MAPK inhibitors, analyzing KEAP1 and NRF2 status in pre-treatment and post-resistance tumor samples will determine if loss of KEAP1 or gain of NRF2 are clinically relevant mechanisms of acquired and intrinsic resistance to these therapies in lung cancer. Stratifying patients based on these findings will be important for evaluating the efficacy of these inhibitors in clinical trials and for choosing the appropriate treatment for patients.

## Materials and methods

### Cell lines and reagents

Cells were obtained from ATCC and fingerprinted as in Barretina et al (*Barretina et al., 2012*). Cells were maintained in RPMI-1640 (NCI-H1299, HCC364, NCI-H1975, and HCC827; Corning, Corning, NY), McCoy's 5A (CALU1; Gibco, Waltham, MA) or DMEM (MGH-065; Invitrogen, Carlsbad, CA) supplemented with 2 mM glutamine, 50 U/mL penicillin, 50 U/mL of streptomycin (Gibco), and 10% fetal bovine serum (Sigma, St. Louis, MO), and incubated at 37°C in 5% $CO_2$. MGH-065 cells were derived as previously described (*Crystal et al., 2014*). Cell lines were tested for mycoplasma prior to screening. Trametinib, vemurafenib, erlotinib, and afatinib, were purchased from Selleck Chemicals (Houston, TX). LDK-378 and Crizotinib were synthesized by the Novartis Global Discovery Chemistry Department.

### Screen optimization for genome scale screens

Blasticidin and puromycin concentrations were optimized for each cell line by treating with different concentrations of drug for 3 days (puromycin) or 7 days (blasticidin). The lowest concentration of drug that killed all cells was used in the screens.

To produce Cas9-expressing cell lines, 200,000–400,000 cells were seeded in one well of a 6-well plate. The following day, cells were infected with 3 mL of pLX311-Cas9 virus with a final concentration of 4 ug/mL polybrene. Cells were spun for 2 hr at 2000 rpm at 30°C. 24 hr after infection, cells were selected with blasticidin for 7 days.

To determine Cas9 activity, parental cell lines and Cas9-expressing cell lines were infected with pXPR_011, a Cas9 activity reporter which expresses eGFP as well as a guide RNA targeting eGFP (*Doench et al., 2014*). 200,000–400,000 cells were seeded in six wells of a 6-well plate and were infected with 25–100 μL virus with a final concentration of 4 μg/mL polybrene. Cells were spun 2 hr at 2000 rpm at 30°C. 24 hr after infection, each well was split into two wells, one of which was selected with puromycin. After 2–3 days of puromycin selection, cells were counted and those with 30–40% infection efficiency were kept for the Cas9 activity assay. After 7 days of puromycin selection, cells were analyzed on an LSRII flow cytometer to determine the amount of GFP-positive cells. Parental cells not expressing Cas9 or pXPR_011 were used as a negative control. Cells expressing pXPR_011 but not Cas9 were used as a positive control.

To optimize inhibitor concentrations, Cas9-expressing cells were infected with different amounts of emptyT virus (to mimic sgRNA infection) and were selected with puromycin. After 3 days of puromycin selection, cells were counted and those with 30–40% infection efficiency were used to optimize inhibitor concentration. Cells were kept in puromycin selection for one week prior to optimizing inhibitor concentration.

To determine the optimal drug concentration for the screens, cells expressing Cas9 and emptyT were treated with different concentrations of drug for three weeks. Cells were passaged or fresh drug-containing media was added every 3–4 days. Cells were counted at each passage. The lowest concentration of drug that resulted in death or proliferative arrest was used in the screen (Extended Data *Figure 1*) In parallel, cells were treated with different concentrations of inhibitor for 24 hr and then lysed in RIPA buffer. Immunoblots were performed with total and phospho-ERK antibodies to determine the concentration of inhibitor that blocked ERK phosphorylation.

To titer the GeCKO v2 library in Cas9-expressing cells, $3 \times 10^{\wedge}6$ cells were seeded per well in a 12-well plate and were infected with different amounts of virus (0, 50, 100, 150, 200, 400 uL), with a final concentration of 4–8 ug/mL polybrene. Cells were spun for 2 hr at 2000 rpm at 30°C. Approximately 6 hr after infection, cells were split into 6-well plates. For each amount of virus, 100,000 cells per well were plated in two wells. 24 hr after infection, one well was treated with puromycin and one

with media alone. After 2–3 days of selection, cells were counted to determine the amount of virus that resulted in 30–40% infection efficiency, and this amount of virus was used in the screen.

## GeCKO v2 library construction

See Sanjana et al (*Sanjana et al., 2014*).

## Genome-scale CRISPR knockout drug resistance screens with GeCKO v2 library

For each screen, two infection replicates were performed. 150 × 10^6 cells were infected per replicate with 40% infection efficiency, in order to obtain 500 cells per sgRNA after selection (60 × 10^6 surviving cells containing 120,000 sgRNAs). 3 × 10^6 cells per well were seeded in 12-well plates and were infected with the amount of virus determined during optimization, with a final polybrene concentration of 4 µg/mL. Plates were spun for 2 hr at 2000 rpm at 30°C. Approximately 6 hr after infection, all wells within a replicate were pooled and were split into T225 flasks. 24 hr after infection, cells were selected in puromycin for one week and were passaged as necessary. After one week of puromycin selection, 60 × 10^6 cells were harvested for the Day 0 time point, and 60 × 10^6 cells were treated with drug. HCC364 cells were treated with 24 nM trametinib or 6.25 µM vemurafenib; H1299 cells were treated with 1.5 µM trametinib; and CALU1 cells were treated with 50 nM trametinib. Cells were passaged or fresh drug-containing media was added every 3–4 days. Drug-treated cells were harvested on Day 14 and Day 21 of drug treatment. To harvest cells, cells were trypsinized, spun down, washed with PBS, and the cell pellets were frozen at −80°C.

Genomic DNA was extracted using the Qiagen Blood and Cell Culture DNA Maxi Kit according to the manufacturer's protocol.

## Sequencing and analysis of genome-scale screens

See Doench et al (*Doench et al., 2016*).

## Screen optimization for focused sgRNA library screen

Live cell time-lapse imaging was used to determine optimal drug concentrations in HCC827 and MGH065 cells. Photomicrographs were taken every 6 hr using an Incucyte live cell imager (Essen Biosciences) and confluence was measured over 130 hr in culture.

## Focused sgRNA library construction

The sgRNA libraries were designed as previously described (*Wang et al., 2014*). A modified tracrRNA scaffold (*Chen et al., 2013*) for Cas9 loading was cloned into the sgRNA vectors before cloning of the guide RNAs. This sgRNA library targets ~2700 genes and is comprised of 20 sgRNAs per gene.

## Focused sgRNA library screens

For the screens in HCC827 and MGH065, cells were infected with a lentiviral sgRNA pool at a representation of 1000 cells per sgRNA at an MOI of 0.5. Cells were selected for four days in the presence of puromycin, and a reference sample was collected 72 hr after selection to ensure adequate selection/representation. Cells were split into a DMSO control arm and a treatment arm. HCC827 cells were treated with 0.01 uM erlotinib and MGH-065 was treated with 0.5 uM Crizotinib or LDK378. Cells were propagated for a total of 14 days with an average sgRNA representation of ≥1000 maintained at each passage. 100 million cells were harvested for DNA extraction by QIAmp Blood Maxi kit (Qiagen), sgRNAs were PCR amplified from 100 ug of genomic DNA, and PCR fragments of 260–280 bp were purified using Agencourt AMpure XP beads (Beckman). The resulting fragments were sequenced on a Hiseq 2500 (Illumina) with a single end 50 bp run. Sequencing reads were aligned to the sgRNA library and the enrichment or loss of individual bar codes or sgRNA were quantified.

## Vectors

Cas9 in the pLX311 backbone (pXPR_BRD111) and sgRNAs in the pXPR_BRD003 backbone were obtained from the Genetic Perturbation Platform at the Broad Institute.

## sgKEAP1 arrayed infection

500,000 cells per well were seeded in 48-well plates in 250 μL media with 4 μg/mL polybrene. 25 μL virus (sgKEAP1 or sgEGFP) was added per well and plates were spun 2 hr at 2000 rpm at 30°C. 6 hr later, each well was split into a 6 cm dish. 24 hr after infection, cells were selected with puromycin for one week.

## TIDE-PCR

We performed TIDE-PCR (*Brinkman et al., 2014*) on CALU1 and HCC364 sgKEAP1 cells (forward primer CTGGTACATGACAGCACCGT and reverse primer TGCTTCACCTACTTTGCAGGA for sgKEAP1–1 cells; forward primer CTCCAGTTTCCTGCCTTGACAT and reverse primer GAACCCCA TGCAGCCAGAT for sgKEAP1–2 cells) and found the editing efficiency to be approximately 93%, of which approximately 70% is a one base pair insertion, which would lead to a frame shift.

## qRT-PCR

RNA was harvested using a Qiagen RNeasy Kit and was reverse transcribed into cDNA using Super-ScriptIII according to the manufacturer's recommendations. KEAP1 mRNA expression was measured using primers downstream of the sgRNA targeting sites. KEAP1 mRNA levels in KEAP1$^{KO}$ cells were ~35–40% of those in control cells.

## Cytoplasmic/nuclear fractionation

$5 \times 10^5$ cells were seeded in 10 cm dishes. The following day, cells were treated with trametinib (25 nM for HCC364 or 50 nM for CALU1) or DMSO. After 72 hr of drug treatment, cells were lysed and fractionated using NE-PER Nuclear and Cytoplasmic Extraction Reagents (Pierce Biotechnology) according to the manufacturer's recommendations.

## Immunoblotting

Cells were lysed in RIPA buffer containing protease and phosphatase inhibitors and were cleared by centrifugation. Protein was quantified using the Pierce BCA assay, and lysate concentrations were normalized. Lysates were run on SDS-PAGE gels and were transferred to nitrocellulose membranes using the Invitrogen iBlot system. Membranes were blocked for one hour in 5% milk in Tris-buffered saline with 0.1% Tween (TBS-T). Membranes were incubated overnight at 4°C with primary antibod-ies in 5% BSA in TBS-T. Membranes were washed three times in TBST-T then incubated 1 hr at room temperature with secondary antibodies in 5% BSA in TBS-T. Membranes were washed in TBS-T and imaged on a Li-Cor Odyssey Infrared Imaging System. Primary antibodies were total ERK (Cell Sig-naling #9102), phospho-ERK (Cell Signaling #4370), total AKT (Cell Signaling #9272), phospho-AKT (Cell Signaling #4060), GAPDH (Cell Signaling #5174), LAMIN A/C (Cell Signaling #4777), KEAP1 (Proteintech 10503–2-AP), and NRF2 (Santa Cruz Biotechnology sc-13032).

## Crystal violet assays

1000–10,000 cells were seeded in 12-well or 24-well plates in the indicated drug conditions. Media containing fresh drug was replaced every 3–4 days. After the indicated number of days, cells were washed in PBS, fixed in 10% formalin for 15 min, and stained with 0.1% crystal violet in 10% ethanol for 20 min. After acquiring images, crystal violet was extracted in 10% acetic acid for 20 min. The absorbance at 565 nm was determined using a Spectramax plate reader.

## ORF expression

293 T cells were seeded in DMEM + 10% FBS + 0.1% Pen/Strep in 6 cm dishes. 24 hr later, cells were transfected with 100 ng VSVG, 900 ng delta8.9, and 1 μg pLX317-ORF plasmid using OptiMEM and Mirus TransIT. 16 hr after transfection, media was changed to DMEM +30% FBS + 1% Pen/ Strep. Virus was harvested 24 hr later. Cell lines were seeded in 6-well plates and were infected the following day with 1:5 dilution of virus containing 4 μg/mL polybrene. 24 hr after infection, cells were selected with puromycin.

## DCFDA assays to measure ROS

Unless otherwise indicated, cells were treated with drug for 3 days. Cells were trypsinized and resuspended in media with 10 μM DCFDA (Sigma D6883) and incubated at 37°C for 90 min in the dark. For a positive control, parental cells were treated with 20 μM tert-butyl hydroperoxide (Sigma Aldrich 458139) during incubation. For a negative control, parental cells were incubated in media without DCFDA. DCFDA fluorescence was detected by flow cytometry, using the FITC channel on an LSRII flow cytometer (BD Biosciences).

## Metabolite profiling

Metabolite extraction was performed as described previously (*Yuan et al., 2012*). In brief, cells were plated in 10 cm dishes and treated for 72 hr with either DMSO or trametinib (25 nM for HCC364 and 50 nM for CALU1) in quadruplicate. Four hours before metabolite extraction, cells were re-fed with fresh media containing drug or DMSO. Cells were then washed with PBS at 37°C, followed by metabolite extraction by scraping in 80% methanol (on dry ice), insoluble materials collected, two sequential extractions combined and stored at −80°C. Prior to mass spectrometry, the metabolite extracts were lyophilized, resolubilized in 20 μL of LC/MS grade water, and then analyzed by liquid chromatography tandem mass spectrometry (LC-MS/MS) using positive ion/negative ion polarity switching via selected reaction monitoring (SRM) on a 5500 QTRAP hybrid triple quadrupole mass spectrometer (AB/SCIEX), as previously described (*Yuan et al., 2012*). Peaks were integrated using MultiQuant 2.1 data analysis software (AB/SCIEX). Metabolite differences were analyzed by normalizing samples by total levels and comparing replicate samples in groups that satisfy significance of $p < 0.05$ using an unpaired two-tailed t-test.

## Xenotransplantation and in vivo drug treatments

Mice were maintained and handled in accordance with the Novartis Institutes for Biomedical Research (NIBR) Animal Care and Use Committee protocols and regulations. Nude mice were inoculated subcutaneously with $10 \times 10^6$ HCC827 KEAP1[KO] or control cells implanted bilaterally in 50% PBS + 50% matrigel. Once the average tumor volume reached 100–200 mm$^3$ in size, mice were randomized to three groups: vehicle, 5 mg/kg erlotinib, or 12 mg/kg erlotinib. Erlotinib was administered by oral gavage once daily for 40 days and tumor volume was measured twice weekly.

## qPCR primer sequences

| Gene | Forward primer | Reverse primer |
| --- | --- | --- |
| KEAP1 | GATTGGCTGTGTGGAGTTGC | GCAGTGGGACAGGTTGAAGA |
| KEAP1 | TCGATGGCCACATCTATGCC | CCGATCCTTCGTGTCAGCAT |
| NFE2L2 | TCCAGTCAGAAACCAGTGGAT | GAATGTCTGCGCCAAAAGCTG |
| GCLC | GTGTTTCCTGGACTGATCCCA | TCCCTCATCCATCTGGCAAC |
| GCLM | CATTTACAGCCTTACTGGGAGG | ATGCAGTCAAATCTGGTGGCA |
| HO1 | CTTTCAGAAGGGCCAGGTGA | GTAGACAGGGGCGAAGACTG |
| NQO1 | CTCACCGAGAGCCTAGTTCC | CGTCCTCTCTGAGTGAGCCA |
| MRP1 | CTCTATCTCTCCCGACATGACC | AGCAGACGATCCACAGCAAAA |
| TKT | GCTGAACCTGAGGAAGATCA | TGTCGAAGTATTTGCCGGTG |
| TALDO1 | GTCATCAACCTGGGAAGGAA | CAACAAATGGGGAGATGAGG |
| PGD | ATATAGGGACACCACAAGACGG | GCATGAGCGATGGGCCATA |
| MTHFD2 | TGTCCTCAACAAAACCAGGG | TTCCTCTGAAATTGAAGCTGG |
| ME1 | CTGCCTGTCATTCTGGATGT | ACCTCTTACTCTTCTCTGCC |
| G6PD | TGACCTGGCCAAGAAGAAGA | CAAAGAAGTCCTCCAGCTTG |
| SHMT1 | TGAACACTGCCATGTGGTGACC | TCTTTGCCAGTCTTGGGATCC |
| SHMT2 | GCCTCATTGACTACAACCAGCTG | ATGTCTGCCAGCAGGTGTGCTT |
| ACTIN | CAACCGCGAGAAGATGACC | ATCACGATGCCAGTGGTACG |

## sgRNA sequences

| Name | Target sequence |
| --- | --- |
| sgGFP | GGCGAGGGCGATGCCACCTA |
| sgKEAP1–1 | CTTGTGGGCCATGAACTGGG |
| sgKEAP1–2 | TGTGTCCTCCACGTCATGAA |
| sgKEAP1–3 | GAGGACACACTTCTCGCCCA |
| sgKEAP1–4 | ACTGGGCGGCCGGTGCATCC |
| sgLACZ-1 | AACGGCGGATTGACCGTAAT |
| sgLACZ-2 | CTAACGCCTGGGTCGAACGC |

## Acknowledgements

We thank Eejung Kim and Kristen Hurov for NRF2 and KEAP1 ORFs, Celina Keating for support and NGS data acquisition, Thales Papagiannakopoulos and Rodrigo Romero for helpful discussions and advice on ROS assays; and Ophir Shalem and Neville Sanjana for advice on screening. We also thank William Sellers and Nicholas Keen for critical discussions. This work was supported in part by US NIH grants: R01 CA130988 (WCH), U01 CA176058 (WCH) and U01 CA199253 (WCH). EBK was supported by postdoctoral fellowships from the Hope Funds for Cancer Research (HFCR-11-03–03) and NIH F32 (CA189306). NI was supported by postdoctoral fellowships from the Susan G Komen Foundation (PDF12230602) and the Terri Brodeur Breast Cancer Foundation. AJA was supported by the Pancreatic Cancer Action Network Samuel Stroum Fellowship, Hope Funds for Cancer Research Postdoctoral Fellowship, American Society of Clinical Oncology Young Investigator Award, Dana-Farber Cancer Institute Hale Center for Pancreatic Cancer, Perry S Levy Endowed Fellowship, and the Harvard Catalyst and Harvard Clinical and Translational Science Center (UL1 TR001102).

## Additional information

### Competing interests

WCH: A consultant and receives research support from Novartis. The other authors declare that no competing interests exist.

### Funding

| Funder | Grant reference number | Author |
| --- | --- | --- |
| National Cancer Institute | R01 CA130998 | William C Hahn |
| National Cancer Institute | U01 CA176058 | William C Hahn |
| National Cancer Institute | U01 CA199253 | William C Hahn |
| Hope Funds for Cancer Research | Postdoctoral Fellowship HFCR-11-03-03 | Elsa B Krall |
| National Institutes of Health | Postdoctoral Fellowship F32 CA189306 | Elsa B Krall |
| Susan G. Komen Foundation | Postdoctoral Fellowship PDF12230602 | Nina Ilic |
| Terri Brodeur Breast Cancer Foundation | Postdoctoral Fellowship | Nina Ilic |
| Pancreatic Cancer Action Network | Samuel Stroum Fellowship | Andrew J Aguirre |

| American Society of Clinical Oncology | Young Investigator Award | Andrew J Aguirre |
| Dana-Farber Cancer Institute Hale Center for Pancreatic Cancer | | Andrew J Aguirre |
| Perry S. Levy Endowed Fellowship | | Andrew J Aguirre |
| Harvard Catalyst and Harvard Clinical and Translational Science Center | UL1 TR001102 | Andrew J Aguirre |

The funders had no role in study design, data collection and interpretation, or the decision to submit the work for publication.

## Author contributions

EBK, Conceptualization, Data curation, Formal analysis, Investigation, Methodology, Writing—original draft, Writing—review and editing, Designed, performed, and analyzed screens, Designed and performed in vitro experiments, Wrote the manuscript; BW, Conceptualization, Data curation, Formal analysis, Investigation, Methodology, Writing—original draft, Writing—review and editing, Performed screens, Designed and performed in vitro experiments, Wrote the manuscript; DMM, Conceptualization, Data curation, Formal analysis, Investigation, Writing—review and editing, Designed, performed, and analyzed screens, Designed and performed in vitro experiments, Designed and performed in vivo experiments, Edited the manuscript; NI, Data curation, Formal analysis, Investigation, Methodology, Writing—review and editing, Designed and performed in vitro experiments, Edited the manuscript; SR, Data curation, Formal analysis, Investigation, Writing—review and editing, Designed and performed in vitro experiments, Edited the manuscript; MJN, Formal analysis, Analyzed patient data, Edited the manuscript; KY, DAR, Data curation, Provided sequencing support and data acquisition; AJA, Methodology, Designed in vitro experiments, Edited the manuscript; JWK, Formal analysis, Performed bioinformatics analyses, Edited the manuscript; AJR, Data curation, Formal analysis, Analyzed patient data; JFG, Data curation, Formal analysis, Obtained patient samples, Edited the manuscript; JAW, Methodology, Designed in vivo experiments; JMA, Data curation, Methodology, Performed mass spectrometry; JGD, Conceptualization, Formal analysis, Methodology, Designed and analyzed screens, Designed in vitro experiments, Edited the manuscript; PAJ, Data curation, Obtained patient samples, Edited the manuscript; ATS, Resources, Obtained patient samples; REMcDI, Supervision, Methodology, Supervised the study, Edited the manuscript; JAE, FS, Supervision, Supervised the study; MRS, Conceptualization, Supervision, Methodology, Designed screens, Designed in vitro experiments, Supervised the study; WCH, Conceptualization, Supervision, Funding acquisition, Writing—original draft, Writing—review and editing, Designed screens, Designed in vitro experiments, Supervised the study, Wrote the manscript

## Author ORCIDs

William C Hahn, http://orcid.org/0000-0003-2840-9791

## Ethics

Animal experimentation: Mice were maintained and handled in accordance with the Novartis Institutes for Biomedical Research (NIBR) Animal Care and Use Committee protocols and regulations.

# Additional files

## Supplementary files

• Supplementary file 1. Genome scale CRISPR-Cas9 screens in CALU1, HCC364, and NCIH1299 cells. For each sgRNA, the average log2 fold-change (LFC) between drug-treated samples and Day 0 samples is shown. The hits from each screen, as determined by LFC or by STARS score, are also summarized.

• Supplementary file 2. Focused sgRNA library screens in HCC827 and MGH065 cells. Raw counts and Z-scores are shown for each sgRNA in DMSO- or drug-treated samples.

• Supplementary file 3. Metabolite Profiling of KEAP1 knockout cells. Metabolite levels are shown for DMSO- or drug-treated samples. Metabolite differences were analyzed by normalizing samples by total levels and comparing replicate samples in groups that satisfy significance of p<0.05 using an unpaired two-tailed t-test.

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
