## [Decision Letter]

Thank you for submitting your article "KEAP1 loss promotes resistance to BRAF, MEK, and EGFR inhibition in lung cancer" for consideration by *eLife*. Your article has been favorably evaluated by Tony Hunter (Senior Editor) and three reviewers, one of whom is a member of our Board of Reviewing Editors. The reviewers have opted to remain anonymous.

The reviewers have discussed the reviews with one another and the Reviewing Editor has drafted this decision to advise you on the additional analysis considered essential to publish this work in *eLife*. At this point, we request that you respond to the comments with a proposed set of experiments that you believe would satisfy the concerns of the reviewers and an estimate of the time it would take to complete this work. The editor and reviewers will consider your response and provide a binding recommendation.

Summary:

The article by Krall et al. describes the results of CRISPR/CAS9 based genetic screening in lung cancer cells to identify mechanisms of resistance to EGFR, BRAF or MEK inhibitor treatment. The authors identify loss of KEAP1 as a mechanism of this resistance. Mechanistically, they link KEAP1 loss to the suppression of reactive oxygen species (ROS) through the de-repression of the KEAP1 target, the transcription factor NRF2, which promotes expression of certain anti-oxidant genes. The authors contend that the induction of ROS is essential for the anti-proliferative effects of EGFR, BRAF or MEK inhibitors. The authors conclude that loss of KEAP1 therefore uncouples MEK inhibition and ROS induction to promote drug resistance. Conceptually, the findings are interesting, novel and potentially provide an important advance in the field, particularly as it pertains to understanding the mechanism and effectiveness of MEK inhibitors in KRAS or BRAF mutated lung cancer. However, there are several important issues that must to be experimentally addressed and/or clarified in the manuscript before this manuscript would be suitable for publication in *eLife*.

Essential revisions:

1) The authors' title indicates that CRISPR/CAS9-mediated KEAP1 silencing promotes resistance to BRAF, MEK, and EGFR inhibition in lung cancer. However, it is not clear whether loss of KEAP1 actually promotes drug resistance, (i.e. the capacity of KEAP(Null) cells to proliferate continuously in the presence of drug in vitro or to sustain tumor growth when transplanted into drug treated mice), or whether KEAP1 silencing simply attenuates the effects of the various inhibitors over the 14-21 day assay period without promoting long-term growth and tumorigenesis per se. For example, the authors' measure of drug resistance documents the effects of either KEAP1 silencing or ectopic NRF2 expression on "Relative Cell Number". CRISPR/CAS9 silencing of KEAP1 leads to a 1.5 to ~4 fold increase in "Relative Cell Number" but it is not clear that this reflects true drug resistance. The authors fail to provide any clonogenicity assays or to provide dose response curves of cell proliferation vs. drug concentration such that GI50 and/or GI90 values may be calculated. Moreover, the authors have not tested whether KEAP1 silencing can promote the growth of xenotransplanted tumors in appropriately drug treated mice, a point that all three reviewers agreed was a key piece of missing data. Hence, because effects of the various manipulations on "Relative Cell Number" were often very modest (from 50% – ~4 fold), there was substantial concern raised by all three reviewers that the authors are potentially overstating their observations with regard to long-term drug resistance vs. short-term attenuation of drug anti-proliferative effects, which substantially undermined enthusiasm for the research as presented and which would have to be addressed directly in any revised manuscript.

2) The authors state that, "We used the lowest concentration of drug that inhibited ERK phosphorylation and resulted in proliferative arrest or death" such that the concentrations of trametinib vary from cell line to cell line. Do the authors believe that different cells have different sensitivity thresholds for trametinib? Moreover, it is not clear that this is the relevant concentration for selection for drug resistance. Indeed, in the trametinib-treated HCC-364 cells there is clearly residual pERK1/2 present in lanes 4-6 of Figure 2. Hence there is concern that the sub-maximal inhibition of BRAF(V600E)->MEK->ERK signaling may allow sufficient signal through this pathway to continue to promote cell proliferation. To that end, to what extent are the sgKEAP1 cells sensitive to an ERK1/2 inhibitor such as SCH772984? An additional concern is that the authors assessed the effects of the various pathway-targeted agents 48 hours post-addition, but their assay for drug resistance was at 14-21 days post drug treatment. Since many cancer cells display adaptive responses leading to ERK1/2 reactivation at late times after drug addition, how did the levels of pERK1/2 compare at 48 hours vs. 14 days post drug-addition? Finally, the authors use A549 cells, which developed as a KRAS mutated, KEAP1-Null cell line. To what extent are A549 cells resistant to trametinib and why is a concentration of 25 nM used in A549 cell experiments but 1.5 microM trametinib was used in KEAP1 proficient H1299 cells?

3) The authors claim to have knocked out KEAP1. What is the significance of the residual KEAP1 bands in Figure 2, and Figure 1—figure supplement 2? Do these represent nonspecific bands, residual expression, degradation products or a sub-population of cells without KEAP1 KO? Is the mRNA expression of KEAP1 undetectable in KO cells? If indeed there is residual KEAP1 expression rather than complete KO, this needs to be addressed in the text. Related to this, the authors demonstrate that trametinib treatment of HCC364 cells leads to decreased KEAP1 and increased NRF2 expression (Figure 3). Indeed, the magnitude of the effects of trametinib on KEAP1 and NRF2 expression appear equivalent to the effects of CRISPR/CAS9-mediated silencing of KEAP1 (compare KEAP1 and NRF2 expression in sgGFP vs. sgKEAP1-1 vs. sgKEAP1-2 in trametinib treated HCC364 cells in Figure 3, lanes 16-18). Hence, if trametinib can promote the expression/activity of NRF2 without the need for CRISPR/CAS9-mediated silencing of KEAP1, should not these cells display some measure of drug resistance?

4) The authors demonstrate that trametinib treatment of HCC364 or CALU1 cells leads to a convincing increase in reactive oxygen species (ROS), which is prevented by CRISPR/CAS9-mediated silencing of KEAP1. They also claim that, "NAC reduced ROS and conferred resistance to trametinib (Figure 5)". The same concern is raised with regard to these experiments as in #1 above: To what extent does NAC influence long-term clonogenicity and influence the dose response of cells to trametinib? In addition, do other MEK inhibitors have a similar effect in KRAS or BRAF mutant models? Does vemurafenib or erlotinib have a similar effect in an appropriate cellular context? Does expression of any NRF2 target genes such as Glutathione Peroxidase (GPX) promote resistance to any of the pathway-targeted agents in the appropriate cellular context? Finally, the effects of KEAP1 knockdown on cell growth and ROS metabolism in the absence of treatment (RAF, MEK inhibitors) is not shown and should be described?

5) Overall, the experiments presented by Krall et al. appear most relevant to primary response/resistance and not to the emergence of acquired resistance. Without any experimental evidence that the KEAP1-NRF2 pathway plays a functional role in acquired resistance, this aspect of the manuscript should be edited to ensure the focus is clearly on the initial treatment context so as not to confuse readers.

6) Access to patient biopsies before or after treatment with pathway-targeted inhibitors (EGFR or MEKi) to determine the presence of KEAP1 alterations would greatly strengthen the impact and clinical relevance of the study. For example, according to the TCGA analysis of lung adenocarcinoma, a percentage of EGFR/ERBB1 mutated lung cancers also carry alterations in either KEAP1 or NRF2. Do the authors have evidence to suggest that patients whose EGFR mutated lung tumors also have alterations in KEAP1 or NRF2 display primary resistance to EGFR blockade? If these are not available, the authors may want to test whether KEAP1 KO attenuates the effect of trametinib (given at MTD) on tumor growth in vivo.

7) The data in the EGFR mutated NSCLC cell lines is underdeveloped and, as it stands, distracts from the bulk of the work in the RAS or BRAF mutated cells.

The full reviews are appended below, in case there are any additional points that you wish to respond to, but this is not mandatory:

*Reviewer #1:*

The authors' title indicates that KEAP1 loss promotes resistance to BRAF, MEK, and EGFR inhibition in lung cancer. However, it is not clear whether loss of KEAP1 actually promotes drug resistance, i.e. the capacity of KEAP(Null) cells to proliferate continuously in the presence of drug in vitro or to form tumors when transplanted into drug treated mice or whether KEAP1 silencing simply provides the cells a short-term survival benefit measured over 14-21 days (Figure 1). For example, the authors' measure of drug resistance in Figure 1 and Figure 2 documents the effects of either KEAP1 silencing or ectopic NRF2 expression on "Relative Cell Number". CRISPR/CAS9 silencing of KEAP1 leads to a 1.5 to ~4 fold increase in Relative Cell Number but it is not clear that this reflects true drug resistance. The authors fail to provide any clonogenicity assays or to provide dose response curves of cell proliferation vs. drug concentration such that GI50 and GI90 values may be calculated. Moreover, the authors have not tested whether KEAP1 silencing can promote the growth of xenotransplanted tumors in drug treated mice. Hence, there is substantial concern that the authors are overstating their observations with regard to long-term drug resistance vs. a short-term survival advantage, which fundamentally undermines the research as presented. Finally, the authors may wish to contrast the fold change of Relative Cell Number following KEAP1 silencing to the effects of a positive control such as the expression of p61-BRAF (V600E), which supports long-term drug resistance to vemurafenib in HCC364 BRAF mutated NSCLC cells.

The effects of NRF2(G31R) over-expression on Relative Cell Number are no greater than 2-fold and in the case of HCC364 cells are ~50% (Figure 2), which is less than that observed with CRISPR/CAS9-mediated silencing of KEAP1. Although the error bars indicate statistical significance, it is not clear that such modest differences would be biologically significant in terms of actual drug resistance per the comment above.

The authors state that, "We used the lowest concentration of drug that inhibited ERK phosphorylation and resulted in proliferative arrest or death". However, it is not clear that this is the relevant concentration for selection for drug resistance. Indeed, in the trametinib treated HCC-364 cells there is clearly residual pERK1/2 present in lanes 4-6 of Figure 2. Hence there is concern that the sub-maximal inhibition of BRAF(V600E)->MEK->ERK signaling may allow sufficient signal through this pathway to continue to promote cell proliferation. To what extent are the sgKEAP1 cells sensitive to an ERK1/2 inhibitor such as SCH772984?

The authors demonstrate that Trametinib treatment of HCC364 cells leads to decreased KEAP1 expression and increased NRF2 expression (Figure 3). Indeed, the magnitude of the effects of trametinib on KEAP1 and NRF2 expression appear equivalent to the effects of CRISPR/CAS9-mediated silencing of KEAP1 (compare KEAP1 and NRF2 expression in sgGFP vs. sgKEAP1-1 vs. sgKEAP1-2 in Trametinib treated HCC364 cells in Figure 3, lanes 16-18). Hence, if trametinib can promote the expression/activity of NRF2 without the need for CRISPR/CAS9-mediated silencing of KEAP1, why are the cells not directly drug resistant?

The authors demonstrate that trametinib treatment of HCC364 or CALU1 cells leads to a convincing increase in reactive oxygen species (ROS), which is prevented by CRISPR/CAS9-mediated silencing of KEAP1. They also claim that, "NAC reduced ROS and conferred resistance to trametinib (Figure 5)". The demonstration of NAC-mediated trametinib resistance was through a purported effect on relative population doublings of trametinib treated vs. trametinib/NAC treated cells (Figure 5). This is not an adequate way in which to document drug resistance, which out to be measured using clonogenicity assays or to provide dose response curves of cell proliferation vs. drug treatment such that effects of NAC on the GI50 and GI90 values may be calculated.

According to the TCGA analysis of lung adenocarcinoma, a percentage of EGFR/ERBB1 mutated lung cancers also carry alterations in either KEAP1 or NRF2. Hence, do the authors have any evidence to suggest that patients whose EGFR mutated lung tumors also have alterations in KEAP1 or NRF2 display primary resistance to EGFR blockade?

*Reviewer #2:*

In this study Krall et al. performed CRISPR/CAS9 screens to determine genes that are required for sensitivity to inhibitors of RTK/RAF/MEK/ERK pathway in lung cancer. They made a very interesting observation that KEAP1 sgRNAs were enriched in screens utilizing trametinib, vemurafenib or erlotinib. They went on to show that KEAP1 KO attenuates the effect of trametinib, vemurafenib and erlotinib in BRAF, KRAS, EGFR models, respectively. From a mechanistic perspective, they show that trametinib induces ROS in cells with wt KEAP1 and, when KEAP1 is lost, increased expression of NRF2 and up regulation of glutathione result in lowered ROS species and decreased viability in the presence of trametinib. Conceptually, the findings are novel and provide an important advance in the field, particularly as it pertains to understanding the effectiveness of MEK inhibitors in KRAS mutant lung cancer. However, there are several issues that need to be experimentally addressed and/or clarified in the manuscript.

1) The authors claim to have knocked out KEAP1. What is the significance of the residual KEAP1 bands in Figure 2, and Figure 1—figure supplement 2? Do these represent nonspecific bands, residual expression, degradation products or subpopulation of cells without KO? Is the mRNA expression of KEAP1 undetectable in KO cells? If indeed there is residual KEAP1 expression rather than complete KO, this needs to be addressed in the text.

2) The authors claim that KEAP1 KO causes resistance to trametinib. How are the authors defining resistance in this study? The data seem to show that KEAP1 KO has an attenuating effect, given that there is a significant antiproliferative effect even in the setting of KEAP1 KO. Also, a single concentration of trametinib was used in most studies. Dose response viability experiments with several trametinib concentrations in the presence or absence of KEAP1 are needed to clarify this issue.

3)KEAP1 sgRNAs scored across different screens/models. One worry is that KEAP1 has a housekeeper function in lung cancer cells. What is the effect of KEAP1 KO on cell viability in the absence of trametinib, vemurafenib or elrotinib treatment? How does this compare to the effect in the presence of trametinib?

4) The concentrations of trametinib vary from cell line to cell line. Do the authors believe that different cells have different sensitivity thresholds for trametinib? The author use A549 cells, which have endogenous loss of KEAP1. Are A549 cells resistant to trametinib? Why is a concentration of 25nM used in A549 but 1.5 μm in H1299?

5) The attenuation of trametinib effect by KEAP1 KO occurs in the absence of ERK reactivation. However, the viability assay was conducted at 21 days, whereas the effect of trametinib on pERK was tested after 48h. What is the status of ERK 21 days after treatment with trametinib in KEAP1 WT vs KO cells? How does this compare to the effect of trametinib 1h after inhibition?

6) The authors claim that resistance occurs because of an increase in NRF2 activity. Overexpression of an NRF2 mutant with defective KEAP1 binding attenuates effect of trametinib and vemurafenib when overexpressed in KEAP1 wt cells. Also, the expression of KEAP1 or a mutant with defective binding to NRF2 in A549 cells results in increased sensitivity to trametinib. These are valid studies, which support the mechanistic conclusions of the manuscript. The magnitude of the effect however is small (e.g. a 10% difference in A549 cells). A more direct approach, i.e. restoring the expression of KEAP1 vs. KEAP1 G333C or knocking out/down NRF2 in KEAP1 KO cells may yield more striking differences.

7) Is the effect of the KEAP1 P128L mutation known? The authors state that KEAP1 KO in these cells increased NRF2 expression. Was the magnitude of such increase similar to the magnitude of the increase observed in other cell lines with WT KEAP1?

8) Trametinib decreases KEAP1 expression, increases NRF2 protein levels and NRF2 dependent gene expression. Why is there an additive effect when combining trametinib with KO (Figure 3)? Could trametinib regulate NRF2 independently of KEAP1?

9) Trametinib induces ROS. Do other MEKis have a similar effect in KRAS or BRAF mutant models? Does vemurafenib or erlotinib have a similar effect in appropriate cellular context?

10) Trametinib induces glutathione in KEAP1 KO cells. Is there an effect on ROS by KEAP1 KO in the absence of treatment (not clear if data in Figure 4 are normalized; there seems to be a small decrease in the supplementary companion).

11) Modulation of ROS by NAC or BSO attenuates or enhances, respectively, the effect of trametinib. In my opinion these are key experiments that bring together the mechanistic insights of the study and could potentially offer hints to optimize the performance of MEK inhibitors in KRAS mutant patients.

Some clarification is needed in the text and figure legends in order to help the reader better understand the experiments/findings.

Why are different concentrations of trametinib used in Figure 5 (50 nM and 10 nM, respectively). Lowering the concentration of trametinib would increase the likelihood that an additive effect on viability is observed. It would be important to show the effect of a fixed concentration of NAC/BSO on the trametinib dose response proliferation curve.

Also, does treatment with NAC or BSO modulate the effect of the other drugs tested in the study?

12) Access to patient biopsies before or after treatment with trametinib to determine the presence of KEAP1 alterations would greatly strengthen the impact and clinical relevance of the study. If these are not available, the authors may want to demonstrate that KEAP1 KO attenuates the effect of trametinib (given at MTD) on tumor growth in vivo.

*Reviewer #3:*

This article describes the results of CRISPR/Cas9 based genetic screening in lung cancer cells to identify mechanisms of resistance to RAF and MEK inhibitor treatment. The authors identify loss of Keap1 as a mechanism of this resistance. Mechanistically, they link Keap1 loss to the suppression of reactive oxygen species (ROS) through the de-repression of the Keap1 target gene NRF2. NRF2 promotes expression of certain anti-oxidant genes. The authors contend that the induction of ROS is important for the ability of MEK inhibition to suppress cell growth. The authors conclude that loss of Keap1 therefore uncouples MEK inhibition and ROS induction to promote resistance.

The studies are interesting, particularly the use of the CRISPR/Cas9 screening in multiple cell lines and the mechanistic connection to ROS and the Keap1-NRF2 axis. However, there are several issues that should be addressed to strengthen the authors' case and fully support their claims. First, the effects of RAF and MEK inhibition on Keap1 and NRF2 levels and function in the cell lines of interest are not shown systematically. Second, the effects of Keap1 knockdown on cell growth and ROS metabolism in the absence of treatment (RAF, MEK inhibitors) is not shown. Third, the data in the EGFR mutant cells is underdeveloped and as it stands distracts from the bulk of the work in the RAS mutant and BRAF mutant cells. Fourth, a limited number of cell lines is used to show the functional importance of either increased or decreased Keap1 and NRF2 in modulating RAF and MEK inhibitor response. This should be expanded. Fifth, no in vivo experiments are shown to corroborate the strength and potential translational relevance of the in vitro findings that are presented. This is especially important to add because most of the in vitro effects on resistance appear relatively modest. Sixth, functional experiments to show that a key metabolism enzyme contributes to the Keap1 knockdown phenotype are absent. Does genetic overexpression of GPX1, for instance, rescue the Keap1 knockdown phenotypes? Finally, the experiments shown deal primarily with primary response/resistance and not acquired resistance. Without any experimental evidence that this Keap1-NRF2 pathway plays a functional role in acquired resistance, this aspect of the manuscript should be edited to ensure the focus is more squarely on the initial treatment context so as not to confuse readers.

---

## [Author Response]

*Essential revisions:*

*1) The authors' title indicates that CRISPR/CAS9-mediated KEAP1 silencing promotes resistance to BRAF, MEK, and EGFR inhibition in lung cancer. However, it is not clear whether loss of KEAP1 actually promotes drug resistance, (i.e. the capacity of KEAP(Null) cells to proliferate continuously in the presence of drug in vitro or to sustain tumor growth when transplanted into drug treated mice), or whether KEAP1 silencing simply attenuates the effects of the various inhibitors over the 14-21 day assay period without promoting long-term growth and tumorigenesis per se. For example, the authors' measure of drug resistance documents the effects of either KEAP1 silencing or ectopic NRF2 expression on "Relative Cell Number". CRISPR/CAS9 silencing of KEAP1 leads to a 1.5 to ~4 fold increase in "Relative Cell Number" but it is not clear that this reflects true drug resistance. The authors fail to provide any clonogenicity assays or to provide dose response curves of cell proliferation vs. drug concentration such that GI50 and/or GI90 values may be calculated. Moreover, the authors have not tested whether KEAP1 silencing can promote the growth of xenotransplanted tumors in appropriately drug treated mice, a point that all three reviewers agreed was a key piece of missing data. Hence, because effects of the various manipulations on "Relative Cell Number" were often very modest (from 50% – ~4 fold), there was substantial concern raised by all three reviewers that the authors are potentially overstating their observations with regard to long-term drug resistance vs. short-term attenuation of drug anti-proliferative effects, which substantially undermined enthusiasm for the research as presented and which would have to be addressed directly in any revised manuscript.*

We agree with the reviewers that the effect observed after deleting KEAP1 is more modest than is sometimes seen when a secondary alteration leads to re-activation of the MAPK pathway to confer drug resistance. However, we believe that our findings are particularly interesting because activation of an independent pathway, rather than the MAPK pathway, is involved.

At the same time, we have performed additional experimentation to support the conclusion that deleting KEAP1 permits cells and tumors to grow in the presence of MAPK pathway inhibitors. Specifically, we now include dose-response curves for HCC827 (EGFR-mutant) sgGFP or sgKEAP1 cells treated with erlotinib (Figure 1—figure supplement 6). We found that the IC50 is indeed shifted in the polyclonal pool of sgKEAP1 cells. Furthermore, when single cell clones with complete KEAP1 knockout were derived, the IC50 of these clones was shifted even further, suggesting that in the initial experiments KEAP1 was not deleted in all of the cells in the population.

In addition, we have now included a xenograft experiment with HCC827 EGFR-mutant cells (Figure 6). HCC827 sgGFP or sgKEAP1 cells were implanted bilaterally into mice, and erlotinib treatment was initiated when the average tumor volume reached 100-200 cm^3^. We show that erlotinib has a cytostatic effect on sgGFP cells, with tumor size remaining stable during drug treatment. In contrast, sgKEAP1 tumors continue to grow larger during erlotinib treatment. Interestingly, sgKEAP1 cells also form larger tumors with vehicle treatment, indicating that loss of KEAP1 also confers a proliferative advantage in the absence of drug in vivo.

Taken together, these new experiments bolster the conclusion that loss of KEAP1 permits cells to proliferate both in vitro and in vivo in the presence of several clinically approved targeted agents, and we thank the reviewers for making these suggestions that have improved the manuscript.

*2) The authors state that, "We used the lowest concentration of drug that inhibited ERK phosphorylation and resulted in proliferative arrest or death" such that the concentrations of trametinib vary from cell line to cell line. Do the authors believe that different cells have different sensitivity thresholds for trametinib? Moreover, it is not clear that this is the relevant concentration for selection for drug resistance. Indeed, in the trametinib-treated HCC-364 cells there is clearly residual pERK1/2 present in lanes 4-6 of Figure 2. Hence there is concern that the sub-maximal inhibition of BRAF(V600E)->MEK->ERK signaling may allow sufficient signal through this pathway to continue to promote cell proliferation. To that end, to what extent are the sgKEAP1 cells sensitive to an ERK1/2 inhibitor such as SCH772984? An additional concern is that the authors assessed the effects of the various pathway-targeted agents 48 hours post-addition, but their assay for drug resistance was at 14-21 days post drug treatment. Since many cancer cells display adaptive responses leading to ERK1/2 reactivation at late times after drug addition, how did the levels of pERK1/2 compare at 48 hours vs. 14 days post drug-addition? Finally, the authors use A549 cells, which developed as a KRAS mutated, KEAP1-Null cell line. To what extent are A549 cells resistant to trametinib and why is a concentration of 25 nM used in A549 cell experiments but 1.5 microM trametinib was used in KEAP1 proficient H1299 cells?*

In our experiments, we found that different cell lines require different concentrations of trametinib to achieve suppression of ERK phosphorylation and proliferative arrest. This likely reflects the differences in driver mutations (NRAS, KRAS, or BRAF) as well as the combination of additional genetic alterations in these different cell lines.

We believe that it is difficult to determine the “relevant” drug concentration for resistance. It remains unclear whether the MAPK pathway is fully or only partially suppressed in human tumors treated with MAPK pathway inhibitors. While the pathway may not be fully inhibited in our experiments, we are comparing cell proliferation with and without KEAP1 loss, and the extent of MAPK pathway inhibition is comparable in these isogenic systems. Therefore, we believe that the differences in proliferation are due to the differences in KEAP1 status, rather than sub-maximal inhibition of the MAPK pathway promoting proliferation.

We have assessed pERK levels at 14 days following drug addition, and the pathway remains suppressed. We have added this experiment as Figure 2—figure supplement 1.

Regarding differing drug sensitivities in different cell lines, H1299 did require a higher concentration of trametinib to cause arrest, although suppression of ERK phosphorylation was observed at a lower concentration. We did not use these cells in any of the validation experiments for this reason. We have analyzed drug sensitivities in a large panel of cell lines (Barretina et al., Nature 2012 and Seashore-Ludlow et al., Cancer Discovery 2015), and found no correlation between KEAP1 status and trametinib sensitivity. This is likely due to the many other genetic and epigenetic differences between these cell lines. We believe our approach of comparing isogenic cell lines with and without KEAP1 (both knocking it out in cells with intact KEAP1 and restoring it in cells that are KEAP1-null) is the best approach to assessing the effect of KEAP1 on drug sensitivity.

*3) The authors claim to have knocked out KEAP1. What is the significance of the residual KEAP1 bands in Figure 2, and Figure 1—figure supplement 2? Do these represent nonspecific bands, residual expression, degradation products or a sub-population of cells without KEAP1 KO? Is the mRNA expression of KEAP1 undetectable in KO cells? If indeed there is residual KEAP1 expression rather than complete KO, this needs to be addressed in the text. Related to this, the authors demonstrate that trametinib treatment of HCC364 cells leads to decreased KEAP1 and increased NRF2 expression (Figure 3). Indeed, the magnitude of the effects of trametinib on KEAP1 and NRF2 expression appear equivalent to the effects of CRISPR/CAS9-mediated silencing of KEAP1 (compare KEAP1 and NRF2 expression in sgGFP vs. sgKEAP1-1 vs. sgKEAP1-2 in trametinib treated HCC364 cells in Figure 3, lanes 16-18). Hence, if trametinib can promote the expression/activity of NRF2 without the need for CRISPR/CAS9-mediated silencing of KEAP1, should not these cells display some measure of drug resistance?*

We believe that the residual bands in Figure 2 and Figure 1—figure supplement 4 represent nonspecific bands. We have performed TIDE-PCR (Brinkman et al., Nucleic Acids Res 2014) on these cells and found the editing efficiency to be approximately 93%, of which approximately 70% is a 1 base pair insertion, which would lead to a frame shift. We also assessed mRNA expression using primers downstream of the sgRNA targeting sites. KEAP1 mRNA levels in KEAP1-KO cells were ~35-45% of that in control cells. This reduction in mRNA levels may reflect nonsense mediated decay of transcripts encoding premature stop codons as a consequence of CRISPR-Cas9 cutting. From the TIDE-PCR results, we expect that most of the remaining transcripts contain frame shift mutations. These experiments have been added to the Methods section of the revised manuscript.

While trametinib decreases KEAP1 expression and increases NRF2 expression, the combination of KEAP1-KO with trametinib treatment causes greater effects on KEAP1 levels, NRF2 levels, and NRF2 target gene expression. We have revised the text to state this observation more clearly (subsection “KEAP1^KO^ confers resistance through increased NRF2 activity”, third paragraph).

*4) The authors demonstrate that trametinib treatment of HCC364 or CALU1 cells leads to a convincing increase in reactive oxygen species (ROS), which is prevented by CRISPR/CAS9-mediated silencing of KEAP1. They also claim that, "NAC reduced ROS and conferred resistance to trametinib (Figure 5)". The same concern is raised with regard to these experiments as in #1 above: To what extent does NAC influence long-term clonogenicity and influence the dose response of cells to trametinib? In addition, do other MEK inhibitors have a similar effect in KRAS or BRAF mutant models? Does vemurafenib or erlotinib have a similar effect in an appropriate cellular context? Does expression of any NRF2 target genes such as Glutathione Peroxidase (GPX) promote resistance to any of the pathway-targeted agents in the appropriate cellular context? Finally, the effects of KEAP1 knockdown on cell growth and ROS metabolism in the absence of treatment (RAF, MEK inhibitors) is not shown and should be described?*

We have found that addition of NAC increases proliferation in the presence of drug, but we do not believe that it fully recapitulates KEAP1 loss. Addition of exogenous NAC to the culture media may not completely mimic the concentration and localization of glutathione that occurs upon KEAP1 loss. We have also shown that KEAP1 loss affects other aspects of cell metabolism, which NAC would not affect.

We believe that KEAP1 loss or overexpression of NRF2 alters expression of many genes, which together affect glutathione synthesis and cellular metabolism. Given the amplitude of the effect that KEAP1 loss or NRF2 overexpression have on proliferation in the presence of drug, we think it is unlikely that overexpressing a single target gene will promote resistance.

As to whether other inhibitors have a similar effect on ROS levels, we have included additional experiments in the revised manuscript showing that the EGFR inhibitor erlotinib and the ALK inhibitor LDK378 induce ROS in EGFR- and ALK-mutant cell lines, respectively, and that this increase is prevented by KEAP1 loss (Figure 4—figure supplement 1).

The effects of KEAP1 knockout on ROS levels in the absence of drug are shown in Figure 4 (compare DMSO-treated sgGFP to DMSO-treated sgKEAP1). KEAP1 loss does not affect cell proliferation in the absence of drug.

*5) Overall, the experiments presented by Krall et al. appear most relevant to primary response/resistance and not to the emergence of acquired resistance. Without any experimental evidence that the KEAP1-NRF2 pathway plays a functional role in acquired resistance, this aspect of the manuscript should be edited to ensure the focus is clearly on the initial treatment context so as not to confuse readers.*

While this manuscript was under revision, Gainor et al. (Cancer Discovery 2016) identified a NRF2 mutation in a patient with acquired resistance to an ALK inhibitor. This mutation (E79Q) is in a mutational hotspot and has previously been shown to impair recognition of NRF2 by KEAP1 (Shibata et al., PNAS 2008), thus activating the pathway. This tumor also had a secondary ALK mutation of unknown function and eventually became resistant to a second generation ALK inhibitor. We believe that the NRF2 mutation may have provided a means for such tumor cells to survive the initial crizotinib treatment and acquire additional resistance mutations over time. We therefore believe that our findings may be relevant for both primary and acquired resistance. We have described this finding in the Discussion section of the revised manuscript and have also clarified that further studies will be necessary to establish the frequency of KEAP1 loss as a resistance mechanism in patients treated with RTK/MAPK pathway inhibitors. We have also modified the title and Abstract to better describe how KEAP1 loss or NRF2 mutations affect the response of cells to these targeted inhibitors.

*6) Access to patient biopsies before or after treatment with pathway-targeted inhibitors (EGFR or MEKi) to determine the presence of KEAP1 alterations would greatly strengthen the impact and clinical relevance of the study. For example, according to the TCGA analysis of lung adenocarcinoma, a percentage of EGFR/ERBB1 mutated lung cancers also carry alterations in either KEAP1 or NRF2. Do the authors have evidence to suggest that patients whose EGFR mutated lung tumors also have alterations in KEAP1 or NRF2 display primary resistance to EGFR blockade? If these are not available, the authors may want to test whether KEAP1 KO attenuates the effect of trametinib (given at MTD) on tumor growth* in vivo.

We agree that it seems reasonable to determine the presence of NRF2/KEAP1 alterations in patient biopsies before and after treatment with targeted inhibitors. Unfortunately, we are limited by the number of available samples. We have looked at patient data with two clinical collaborators who are leaders in leading such trials and have collected and followed different cohorts of patients as well as with a major pharmaceutical company. Despite these joint studies, the number of patient samples available for these analyses are quite small and do not preclude that such mutations occur in a fraction of treated patients.

We note (see response to comment #5), that Gainor et al., recently identified a patient that harbors a NRF2 mutation in a drug-resistant tumor but not the pre-treatment tumor. As the reviewers commented above, we believe our results may also be relevant to primary/intrinsic resistance. Although RTK/Ras alterations can co-occur with NRF2/KEAP1 alterations, the number of such patients is small, as is the number of patients enrolled in clinical trials. Thus, there is not sufficient statistical power to assess for a correlation between NRF2/KEAP1 status and drug response. At this point, a sufficient number of patients have not yet been studied to assess this point directly. As more patients are treated with RTK/MAPK pathway inhibitors, it will be extremely interesting to look for such a correlation.

As described above, the revised manuscript now includes an in vivo tumor xenograft experiment (Figure 6), showing that EGFR-mutant tumors with intact KEAP1 fail to grow in the presence of erlotinib treatment, while KEAP1-KO tumors continue to grow larger during treatment.

*7) The data in the EGFR mutated NSCLC cell lines is underdeveloped and, as it stands, distracts from the bulk of the work in the RAS or BRAF mutated cells.*

While our original submitted manuscript focused on MEK inhibition in KRAS and BRAF mutant cells, we subsequently found that KEAP1 loss also conferred resistance to EGFR and ALK inhibition. We believe the observation that KEAP1 loss confers resistance to multiple targeted therapies in lung cancer is a very important finding. We have added additional CRISPR-Cas9 screens showing that KEAP1 loss confers resistance to EGFR and ALK inhibitors, and that it does so by a similar mechanism to the one identified in trametinib-treated KRAS and BRAF mutant cells.